# A fungal NRPS-PKS enzyme catalyses the formation of the flavonoid naringenin

Hongjiao Zhang[1,2,5], Zixin Li[1,2,5], Shuang Zhou[1,2], Shu-Ming Li [3], Huomiao Ran[1], Zili Song[1,2], Tao Yu [4] & Wen-Bing Yin [1,2] ✉

Biosynthesis of the flavonoid naringenin in plants and bacteria is commonly catalysed by a type III polyketide synthase (PKS) using one *p*-coumaroyl-CoA and three malonyl-CoA molecules as substrates. Here, we report a fungal non-ribosomal peptide synthetase -polyketide synthase (NRPS-PKS) hybrid FnsA for the naringenin formation. Feeding experiments with isotope-labelled precursors demonstrate that FnsA accepts not only *p*-coumaric acid (*p*-CA), but also *p*-hydroxybenzoic acid (*p*-HBA) as starter units, with three or four malonyl-CoA molecules for elongation, respectively. In vitro assays and MS/MS analysis prove that both *p*-CA and *p*-HBA are firstly activated by the adenylation domain of FnsA. Phylogenetic analysis reveals that the PKS portion of FnsA shares high sequence homology with type I PKSs. Refactoring the biosynthetic pathway in yeast with the involvement of *fnsA* provides an alternative approach for the production of flavonoids such as isorhamnetin and acacetin.

Flavonoids are an important class of phenolic compounds widely distributed in nature and have been shown to possess antioxidant, antiproliferative, anti-inflammatory and anti-diabetic properties[1,2]. This makes them popular molecules in the pharmaceutical and nutraceutical industries[3]. To date, more than 2500 flavonoid structures have been discovered[4], and more than 95% of them are from plants. However, the low product yields in plants have become the bottleneck for applications. Although flavonoids are sporadically found in microorganisms, bacteria and fungi are often used as hosts for their production[5–7].

Naringenin and naringenin chalcone are the most abundant members and building blocks of diverse flavonoids[8]. As the key precursor of flavonoids, the formation of naringenin has been extensively studied in plants[9] and bacteria[10]. Generally, the naringenin biosynthesis starts with ʟ-tyrosine, which is converted to *p*-coumaric acid (*p*-CA) by tyrosine ammonia lyase. *p*-CA is subsequently converted by 4-coumaroyl-CoA ligase (4CL) to *p*-coumaroyl-CoA. Then, a type III PKS, i.e., the chalcone synthase (CHS), catalyses the condensation of one molecule of *p*-coumaroyl-CoA with three molecules of malonyl-CoA

under the elimination of four molecules of HS-CoA and three molecules of $CO_2$. The CHS product naringenin chalcone is converted to naringenin by chalcone isomerase (CHI) or by non-enzymatic cyclisation under alkaline conditions[11]. In total, two or three enzyme-catalysed reaction steps are required for naringenin formation from *p*-CA. Recently, this common naringenin biosynthetic pathway has been engineered into microorganisms to generate valuable flavonoids such as scutellarein[12], breviscapine[13], and icariin[14] in bacteria or yeasts. It is worth mentioning that fungi can produce various flavonoids such as naringenin, quercetin, kaempferol, and chlorflavonin[15,16]. Notably, only two examples of 4CL- and CHS-like proteins identified from the endophytic fungi *Alternaria* sp. MG1[17] and *Phomopsis liquidambaris*[15] have been reported to catalyse formation of naringenin and its derivatives. The biosynthesis of fungal-derived flavonoids is still rarely studied.

In this work, we identify a fungal NRPS-PKS hybrid enzyme harbouring a unique domain architecture, termed as FnsA, from a plant endophytic fungus *Pestalotiopsis fici* CGMCC3.15140 (*P. fici*). The functional evidence of FnsA as an unexplored fungal naringenin synthetase is provided through heterologous expression, feeding

[1]State Key Laboratory of Mycology, Institute of Microbiology, Chinese Academy of Sciences, Beijing 100101, China. [2]Savaid Medical School, University of Chinese Academy of Sciences, Beijing 100049, China. [3]Institut für Pharmazeutische Biologie und Biotechnologie, Fachbereich Pharmazie, Philipps-Universität Marburg, Marburg 35037, Germany. [4]Center for Synthetic Biochemistry, CAS Key Laboratory of Quantitative Engineering Biology, Shenzhen Institute of Synthetic Biology, Shenzhen Institutes for Advanced Technology, Chinese Academy of Sciences, Shenzhen 518055, China. [5]These authors contributed equally: Hongjiao Zhang, Zixin Li. ✉e-mail: yinwb@im.ac.cn

experiments and in vitro biochemical assays. FnsA catalyses the formation of naringenin with either *p*-CA or *p*-hydroxybenzoic acid (*p*-HBA) as substrate on a single protein, differing from the conventional plant naringenin pathway described previously. Using FnsA, we construct de novo biosynthesis of plant-derived flavonoids, i.e., isorhamnetin and acacetin, in the engineered *fnsA*-containing *Saccharomyces cerevisiae*. Our study provides an alternative engineering approach to the key precursor naringenin for the efficient production of plant flavonoids in yeast.

## Results

### Identification of a fungal NRPS-PKS hybrid enzyme FnsA

The enzyme family of fungal NRPS-PKS has raised more and more attention due to their sometimes cryptic catalytic function. For example, a NRPS-PKS hybrid (P168DRAFT_323099) from *Aspergillus campestris* has been proposed to be involved in the formation of the flavonoid chlorflavonin[16]. To explore the function of this putative biosynthetic enzyme showing a unique domain architecture, we searched over 2000 public fungal genomes and found 502 fungal NRPS-PKS hybrid enzymes. Among them, the catalytic functions of only five enzymes are known (TAS1 for tenuazonic acid[18], SwnK for swainsonine[19], HispS for hispidin[20], AnATPKS for pyrophen[21], and HppS for 4-hydroxy-6-(4-hydroxyphenyl)-*α*-pyrone[22] formation), indicating that more than 99% of the enzymes remain to be explored (Fig. 1a and Supplementary Data 1). We focused on the secondary metabolite producing fungus *P. fici* and identified a NRPS-PKS hybrid enzyme PFICI_04360 with the domain architecture A-T-KS-AT-DH-KR-ACP-TE (A, adenylation; T, thiolation; KS, ketosynthase; AT, acyltransferase; DH, dehydratase; KR, ketoacyl reductase; ACP, acyl carrier protein; TE, thioesterase) (Fig. 1b), and sharing a sequence identity of 64.3% with P168DRAFT_323099 on the amino acid level. In-depth analysis indicated that the upstream and downstream sequences of *PFICI_04360* are likely not related genes for the biosynthesis (Supplementary Table 1), suggesting *PFICI_04360* as an independently functional gene.

To address the function of *PFICI_04360*, we deleted the gene in *P. fici* (Supplementary Fig. 1). High-performance liquid chromatography (HPLC) analysis of the ethyl acetate (EtOAc) extracts of cultures

revealed no obvious differences between wild type and the deletion strain (Fig. 1c), indicating that *PFICI_04360* was silent or lowly expressed. This result is consistent with the previously reported transcriptome data[23]. To activate this gene, we expressed it in the heterologous host *Aspergillus nidulans* LO8030 under the control of the constitutive glyceraldehyde-3-phosphate dehydrogenase (*gpdA*) promoter[24] (Supplementary Fig. 2). As shown in Fig. 1d, overexpression of *PFICI_04360* led to the accumulation of two additional products (**1** and **2**) in comparison to the control strain after nine-day cultivation (Fig. 1d). Further liquid chromatography-mass spectrometry (LC-MS) detection proved the [M + H]$^+$ ion of **1** to be *m/z* 273.0761 and that of **2** at *m/z* 273.0757. Isolation and structure elucidation (Supplementary Figs. 16–19) confirmed **1** and **2** as naringenin and naringenin chalcone, respectively. The optical rotation of **1** was determined at −19.23° and thus identified as (*S*)-naringenin[25]. Taken together, the NRPS-PKS hybrid enzyme encoded by *PFICI_04360* generates naringenin and naringenin chalcone following an unidentified biosynthetic logic. Therefore, it is an unexplored type of fungal naringenin synthase, which is termed FnsA hereafter.

### Functional characterization of *fnsA*

To confirm the function of FnsA, we cloned *fnsA* gene from the complementary DNA (cDNA) and expressed it under the control of *ADH2p* promoter in *Saccharomyces cerevisiae* BJ5464-NpgA (Supplementary Fig. 3), which contains a *holo*-ACP synthase NpgA for PKS- and NRPS-modification[26]. HPLC analysis indicated the presence of trace amounts of **1** in the *fnsA* expressing strain (Fig. 2a). This proved the function of FnsA and suggested insufficient supply of precursors in *S. cerevisiae*. Considering the reported biosynthetic pathway of naringenin and the substrate promiscuity of the A domains in NRPS-PKS hybrids[21], we proposed that FnsA might recognize *p*-CA (**3**) or simple molecules such as free or protein-bound amino acids as substrates for the biosynthesis of **1**. To confirm the hypothesis, **3**, ʟ-phenylalanine, ʟ-tyrosine, cinnamic acid, caffeic acid, ferulic acid, *p*-HBA (**4**), salicylic acid, and anthranilic acid were fed to *S. cerevisiae* containing *fnsA*, respectively. Interestingly, significant accumulation of **1** was only observed in the culture extracts of transformants supplemented with **3** or **4** (Fig. 2b). In contrast, no

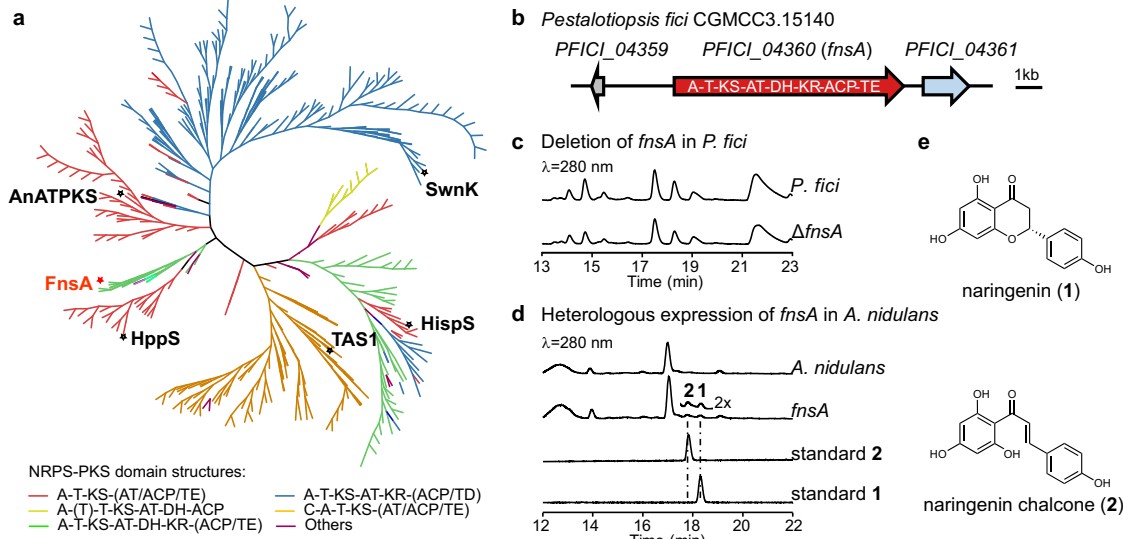

**Fig. 1 | Identification of a fungal NRPS-PKS hybrid enzyme FnsA (PFICI_04360).** **a** Phylogenetic analysis of NRPS-PKS hybrid enzymes from fungi. Fungal NRPS-PKS hybrid enzymes with various domain structures are widely spread in the fungal kingdom. TAS1[18], SwnK[19], HispS[20], AnATPKS[21], and HppS[22] were the identified fungal NRPS-PKS hybrid enzymes. The types of domain structures were shown in different colours. Domains in parenthesis are flexible. A adenylation, T thiolation, KS ketosynthase, AT acyltransferase, DH dehydratase, KR ketoacyl reductase, ACP acyl

carrier protein, TE thioesterase. **b** Portion of the putative *fnsA*-containing gene cluster from *P. fici*. PFICI_04360 is a NRPS-PKS hybrid enzyme with domain architecture indicated. **c** HPLC analysis of secondary metabolites of *P. fici* and the *fnsA*-knockout mutant. **d** HPLC analysis of secondary metabolites of *A. nidulans* and the transformant overexpressing *fnsA*. **e** Structures of the identified products **1** and **2**. Source data are provided as a Source Data file.

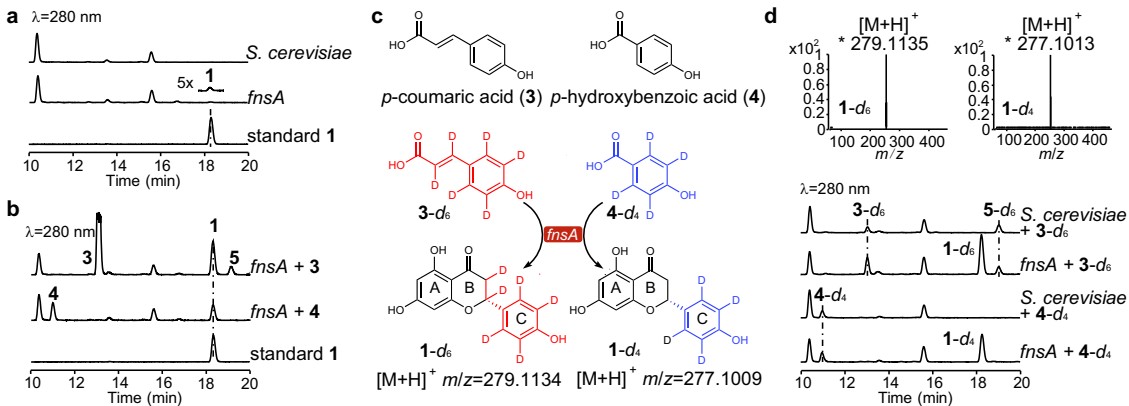

**Fig. 2 | Feeding study in *S. cerevisiae* strain carrying *fnsA*. a** HPLC analysis of secondary metabolites of *S. cerevisiae* and the transformant containing *fnsA*. **b** HPLC analysis of secondary metabolites of *S. cerevisiae* containing *fnsA* after feeding experiment. **c** Structures of the precursors **3** and **4** and the proposed reactions of FnsA with the isotope-labelled precursor **3**-$d_6$ or **4**-$d_4$. **d** HPLC analysis of secondary metabolites of *S. cerevisiae* containing *fnsA* after feeding with **3**-$d_6$ or **4**-$d_4$ and the isotopic patterns of [M + H]$^+$ ions of the labelled products **1**-$d_6$ and **1**-$d_4$. Source data are provided as a Source Data file.

difference was detected between extracts without addition or with other substrates (Supplementary Fig. 4). Moreover, an additional product (**5**) with a [M + H]$^+$ ion at $m/z$ 121.0649 was detected when feeding **3**. Compound **5** was purified from a large-scale fermentation and characterized as 4-hydroxystyrene by comparison with the reported [1]H and [13]C NMR data (Supplementary Figs. 20 and 21). Conversion of **3** to **5** was reported by phenylacrylic acid decarboxylase and ferulic acid decarboxylase in *S. cerevisiae*[27] (Supplementary Fig. 5). To determine whether production of **2** is due to the spontaneous conversion of **1** in *S. cerevisiae* carrying *fnsA*, samples from the feeding experiment were analysed by HPLC at different time points (6, 12, 24, and 48 h). Accumulation of **1** and **2** was detected after 6 h cultivation after feeding with **3** or **4** in *fnsA*-expressing *S. cerevisiae*. Conversion of **2** to **1** was observed afterward until **2** was totally transformed to **1** after 48 h cultivation (Supplementary Fig. 6). These outcomes indicate that FnsA accepts **3** or **4** to synthesize **1** and **2** in vivo. Given the structures of naringenin, **3** or **4** very likely constitutes the C ring of **1** (Fig. 2c). The feeding experiments proceeded with isotope-labelled **3** or **4**, i.e., **3**-$d_6$ or **4**-$d_4$. The dominant isotopic peaks of the [M + H]$^+$ ions of the products after **3**-$d_6$ and **4**-$d_4$ feeding were shifted from $m/z$ 273.0758 to 279.1135 and 277.1013, respectively, which are 6 and 4 Da larger than the natural abundance of **1** (Figs. 2c and 2d). These results confirm that *p*-CA or *p*-HBA serves as the precursors for **1** biosynthesis.

## The FnsA A domain is essential for substrate recognition and activation

We speculated that the A domain of FnsA recognizes and activates the carboxyl group of **3** or **4** in an acyl-*O*-AMP form, and subsequently becomes covalently attached to the cognate T domain, resulting in the formation of an acyl thioester. The PKS portion of FnsA would then construct the ketide chain using malonyl-CoA. To explore the function of the FnsA A domain, the boundary between the NPRS (*fnsA*$^{A-T}$) and PKS portions (*fnsA*$^{PKS}$) were defined and the cDNA sequences of *fnsA*$^{A-T}$ and *fnsA*$^{PKS}$ were cloned and expressed in *S. cerevisiae*. HPLC analysis of the culture extracts revealed that only *S. cerevisiae* harbouring both *fnsA*$^{A-T}$ and *fnsA*$^{PKS}$ could produce **1** but not strains with *fnsA*$^{A-T}$ or *fnsA*$^{PKS}$ alone (Fig. 3a, b). These results suggested that FnsA$^{A-T}$ is required for the activation of substrates **3** or **4**. For further confirmation, we prepared FnsA recombinantly and performed in vitro enzymatic reactions in the presence of **3** or **4**. Two products with [M + H]$^+$ ion of **6** at $m/z$ 494.1072 and that of **7** at $m/z$ 468.0917 were detected by LC-MS analysis and indicated the formation of **3**-AMP (**6**) and **4**-AMP (**7**), respectively (Fig. 3c–e). This result suggested that the FnsA A domain activated **3** or **4** as adenylates, followed by transfer to the thiol group of the pantetheinyl residue of the T domain (Fig. 4a), which were subsequently transferred to the adjacent PKS portion of FnsA. To investigate the catalytic function of the PKS portion, we prepared FnsA$^{PKS}$ recombinantly and performed in vitro enzymatic reactions in the presence of the pre-activated CoA ester of **3**, *p*-coumaroyl-CoA. HPLC analysis of the reaction extracts showed no formation of **1** (Supplementary Fig. 7). Therefore, it is likely that *p*-CA is only accepted by the PKS portion when it is covalently attached to the T domain of FnsA. *p*-Coumaroyl-CoA can be used neither by its A domain nor the KS domain as substrate. This hypothesis is consistent with the results of the phylogenetic analysis of the FnsA KS domain, which is far away from type III PKSs using *p*-coumaroyl-CoA as substrate (Fig. 5a).

To determine the function of the FnsA A domain, recombinant *N*-terminally *His₆*-tagged *holo*-FnsA$^{A-T}$ was purified from *Escherichia coli* BAP1 and incubated with **3** or **4** at 25 °C. The 10 h reaction mixtures were loaded onto SDS-PAGE and the substrate-bound FnsA$^{A-T}$ fragments migrated to a position corresponding to a molecular weight between 70 and 80 kDa. The targeted fragments were collected and subjected to brief proteolytic digestion with trypsin, resulting in the formation of peptide fragment mixtures ranging from 144 to 4944 Da for LC-MS/MS analysis (Supplementary Tables 3 and 4). The enzyme reaction of FnsA$^{A-T}$ with **3** led to a 486.1220 Da increase at Ser607 of the targeted peptide (YADESFSHLGLT**S**MAGVVLR) in comparison to that of the *apo*-FnsA$^{A-T}$ (Fig. 4b, c, and Supplementary Table 3), confirming that **3** is covalently bound to the thiol group of the phosphopantetheinyl residue in the T domain of FnsA$^{A-T}$. Likewise, the attachment of **4** was detected with a mass increase of 444.1115 Da (additional 15.9955 Da due to the Met608-oxidation from the thioether to the sulfoxide) at Ser607 by LC-MS/MS analysis (Fig. 4b, c and Supplementary Table 4). The mass shifts observed were identical to the theoretical values. These results confirmed that the FnsA A domain is responsible for the adenylation and thioester formation of **3** or **4** as well as the covalent binding to the thiol group of the phosphopantetheinyl residue of the FnsA T domain. Next, we tested the substrate specificity of the FnsA A domain. The results showed that besides **3** and **4**, the FnsA A domain could activate other substrates such as cinnamic acid and salicylic acid (Supplementary Fig. 8). Feeding experiments with different concentration of **3** or **4** determined that FnsA prefers **3** as substrate (Supplementary Fig. 9). All these results proved that the FnsA A domain is essential for the recognition and activation of *p*-CA or *p*-HBA in the biosynthesis of naringenin.

## FnsA is an unexplored naringenin synthase

CHS is a well-known ubiquitous type III PKS for the biosynthesis of naringenin. Generally, CHS uses already CoA-activated units as

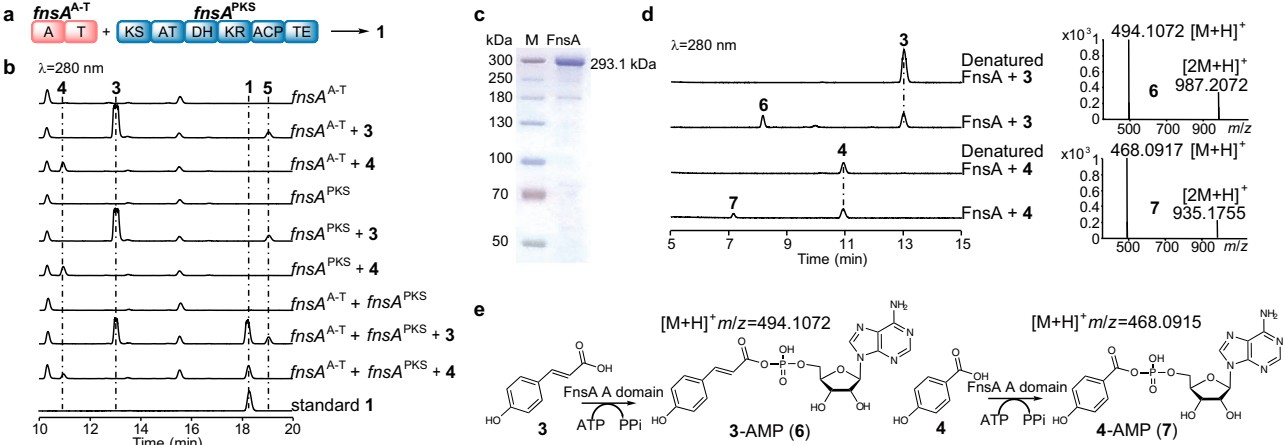

**Fig. 3 | The FnsA A domain is essential for the recognition and activation of *p*-CA (3) or *p*-HBA (4). a** Schematic representation of the NRPS and PKS portions of FnsA. **b** HPLC analysis of secondary metabolites of *S. cerevisiae* strain carrying *fnsA*[A-T], *fnsA*[PKS], and both portions of FnsA in the feeding experiment. **c** SDS-PAGE analysis of the purified FnsA. M, marker, the yield of FnsA protein was 1 mg L[−1].

**d** HPLC analysis of the enzyme reactions of FnsA with **3** or **4**. EICs at *m/z* 494.1072 and 468.0917 refer to [M + H]+ ions of products **3**-AMP (**6**) and **4**-AMP (**7**), respectively. **e** Illustration of the reactions catalysed by the FnsA A domain. Each experiment (**c**) was repeated twice independently with similar results. Source data are provided as a Source Data file.

substrates for the multiple-step biosynthesis of naringenin. However, the fungal NRPS-PKS hybrid FnsA uses free aryl acids, *p*-CA or *p*-HBA, and malonyl-CoA for the formation of naringenin. This difference aroused curiosity about the function of the PKS portion of FnsA. Therefore, we performed a phylogenetic analysis of the KS domains of type I, II, and III PKSs as well as those of FnsA homologues. As shown in Fig. 5a and Supplementary Fig. 10, the KS domains of FnsA and its homologues were classified into an independent clade and close to that of the type I PKSs.

Based on the results described above, we propose the biosynthetic steps to naringenin in *P. fici* (Fig. 5b). *p*-CA (**3**) or *p*-HBA (**4**) acts as starter unit, which is recognized by the A domain of FnsA and loaded onto the adjacent T domain as acyl thioester. The activated **3** or **4** is subsequently elongated with three or four malonyl-CoA units by the AT and KS domains of FnsA. Afterwards, naringenin chalcone is cyclized through Claisen condensation and thereby released either spontaneously or catalysed by the TE domain. Finally, naringenin chalcone is converted to naringenin spontaneously or by a CHI.

## De novo biosynthesis and refactoring of flavonoid pathways in engineered yeast cells

After confirmation of the FnsA function, we engineered heterologous production of naringenin in yeast by combining genes encoding the biosynthetic pathways for its substrates **3** or **4** and *fnsA*. According to previous report, we overexpressed chorismate lyase gene *ubiC*, shikimate kinase gene *aroL* and deleted the endogenous genes for chorismate mutase (*ARO7*) and indole-3-glycerol-phosphate synthase (*TRP3*) which block the production of **4** in *S. cerevisiae* BJ5464-NpgA[28] (Supplementary Fig. 11). This resulted in a product yield to 192.8 mg L[−1] of **4** (Supplementary Fig. 12). To achieve production of naringenin, we introduced *fnsA* into the **4**-producing yeast strain (TYHJ18). The resulting strain produced naringenin at a titer of 10.3 mg L[−1] (Fig. 6b). Alternatively, the engineered *S. cerevisiae* strain QL35[29] showing high production of **3** at about 2 g L[−1] was used as host strain for better substrate supply. *fnsA* from *P. fici* and *npgA* coding for a *holo*-ACP synthase from *A. nidulans* were transformed into QL35 and the resulting strain TYHJ45, termed as QL35-NAR hereafter, produced naringenin at 30.2 mg L[−1] (Fig. 6b).

As a proof of concept, we performed de novo biosynthesis of biologically active flavonoids, *i.e.*, isorhamnetin (**8**) and acacetin (**9**) based on the engineered yeast strain QL35-NAR for naringenin production. To achieve isorhamnetin production, we introduced four

biosynthetic genes including the flavonol synthase gene *PdFLS*[30] from *Populus deltoides*, the flavanone 3-hydroxylase gene *AtF3H*[30], the flavonol 3′-hydroxylase gene *AtF3′H*[31] and the *O*-methyltransferase 1 gene *AtCOMT1*[32] from *Arabidopsis thaliana*, into QL35-NAR to construct the **8**-producing strain TYHJ48, termed as QL35-ISO hereafter, (Fig. 6a and Supplementary Fig. 13). This led to the production of **8** at 1.5 mg L[−1] after cultivation in YPD medium culture for 96 h (Fig. 6c, Supplementary Figs. 14 and 15). In addition, **9**-producing strain TYHJ49, termed as QL35-ACA hereafter, was constructed by introducing the flavone synthase I gene *EbFNSI*[13] from *Erigeron breviscapus* and the flavone 4′-*O*-methyltransferase gene *PaCOMT*[33] from *Plagiochasma appendiculatum* into QL35-NAR (Fig. 6a and Supplementary Fig. 13). We detected **9** production at 7.8 mg L[−1] after fermentation in 100 mL of YPD medium (Fig. 6c, Supplementary Figs. 14 and 15). To improve the production of these flavonoids, we optimized aeration which is known to strongly affect product titers in microbial fermentation[34]. The engineered strains were therefore cultivated in shake flasks with less medium to improve the aeration. Under the optimized conditions, the yield of **8** was increased 2.1-fold to 3.1 mg L[−1] (Fig. 6d) and the production of **9** was improved from 7.8 to 10.4 mg L[−1] after 96 h cultivation (Fig. 6e). In this way, we successfully de novo synthesized two bioactive flavonoids, isorhamnetin and acacetin via the naringenin synthase FnsA.

## Discussion

Naringenin is the key precursor for the biosynthesis of bioactive plant flavonoids. It has attracted great attention for its biosynthesis and product efficiency[8,35,36]. In order to improve naringenin production, researchers adopted various strategies, such as optimizing pathway genes[37], modular pathway engineering[38], and increasing the supply of precursors[38]. It is well known that naringenin biosynthesis in plants and bacteria requires two key enzymes, 4CL and CHS, to form the molecular skeleton of flavonoids. In addition to plants, a number of flavonoids have been isolated from fungi. However, little is known about the genes or pathways involved in their biosynthesis. Herein, we discovered a NRPS-PKS hybrid enzyme FnsA from the endophytic fungus *P. fici* by targeted genome mining. Heterologous expression of *fnsA* in *A. nidulans* and *S. cerevisiae* confirmed its function as a naringenin synthase. Based on the chemical structure of naringenin and the known biosynthesis in plants and bacteria, the free aromatic acids *p*-CA and *p*-HBA were considered as biosynthetic precursors of naringenin, which was confirmed by in vivo feeding experiments (Fig. 2d). In addition, compared with the biosynthesis of naringenin chalcone

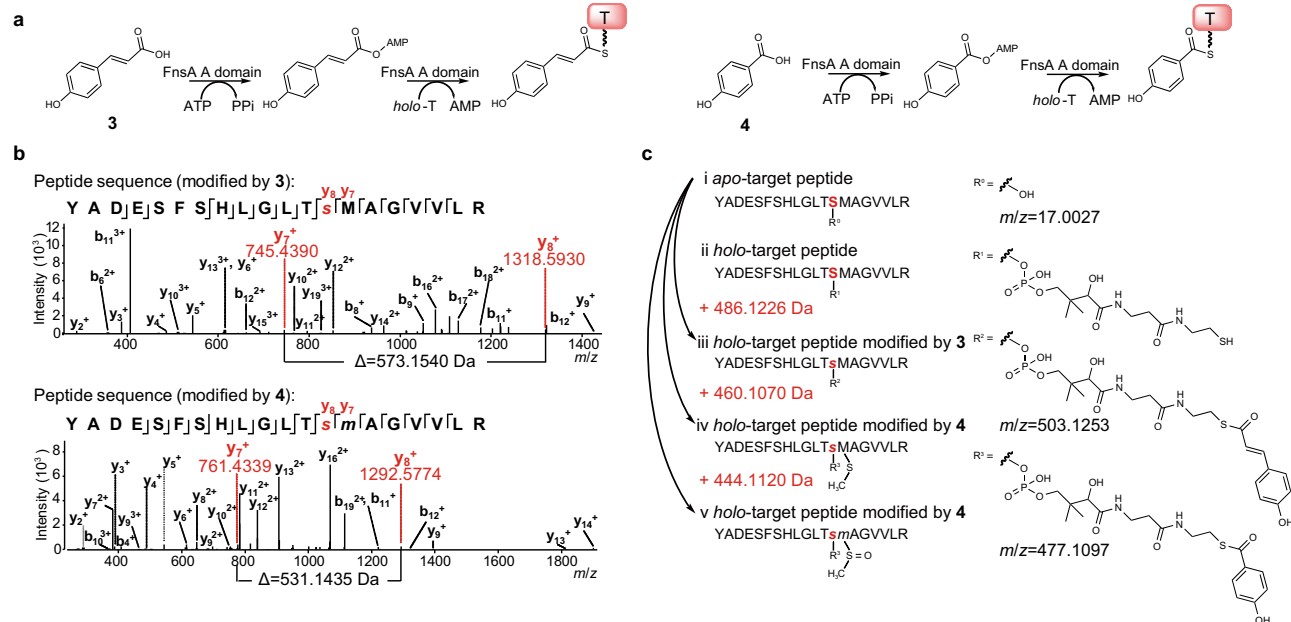

**Fig. 4 | The FnsA A domain is responsible for the adenylation and thioesterification of 3 or 4. a** Illustration of the reactions catalysed by the FnsA A domain when incubating with **3** or **4**. **b** LC-MS/MS fragmentations for the target peptide of FnsA[A-T] covalently attached to **3** or **4** after in vitro assays. **c** Structures and theoretical masses shifts of the molecules that loaded **3** or **4** on Ser607 (marked in red). The labels "b" and "y" assign the *N*- and *C*-terminal fragment ions of the peptide produced by collision-induced fragmentation at the peptide bond in the mass spectrometer. The subscripted number (e.g., $y_7$, $y_8$) represents the number of *N*- or *C*-terminal residues present in the peptide fragment. The mass of Serine residue is 87.0320 Da. Compared to *apo*-target peptide, Ser607 modified by **3** is with a theoretical mass shift of +486.1226 Da and Ser607 modified by **4** is with theoretical mass shifts of +460.1070 Da and +444.1120 Da when Met608 is oxidized.

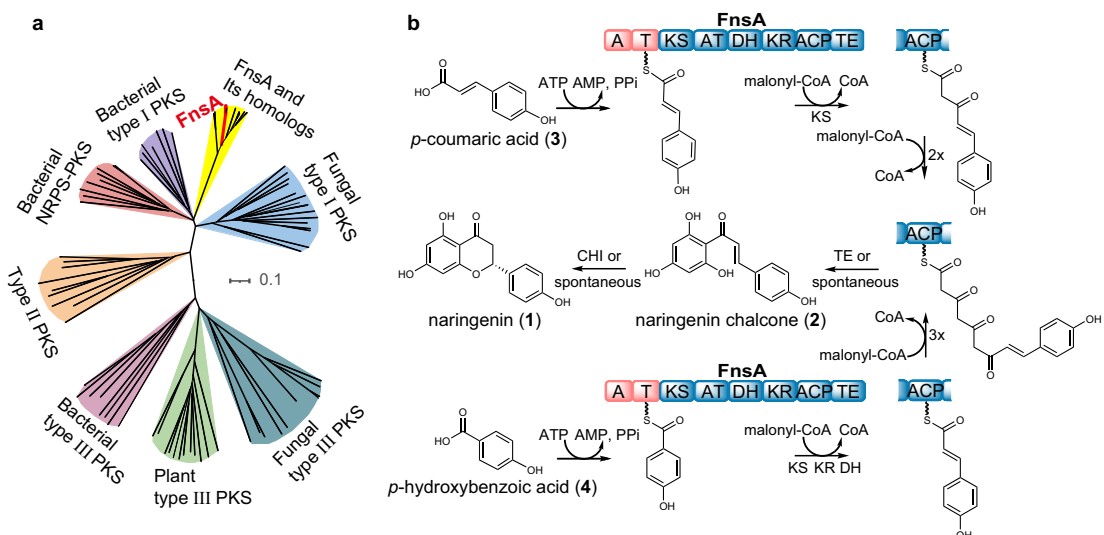

**Fig. 5 | Phylogenetic analysis of the FnsA KS domain and the proposed biosynthetic pathway of naringenin in *P. fici*. a** Phylogenetic analysis of the FnsA KS domain and functional characterized KS domains from plants, bacteria, and fungi. FnsA is highlighted in red. The KS domains homologous to FnsA are indicated in yellow. Scale bar, 0.1 substitutions per site. **b** Proposed formation mechanism of naringenin in *P. fici*. CHI, chalcone isomerase.

catalysed by 4CL and CHS in plants, FnsA can synthesize naringenin alone by using *p*-CA with the involvement of three malonyl-CoA molecules. It should be emphasized that FnsA can also use *p*-HBA as a substrate for the biosynthesis of naringenin chalcone with the involvement of four malonyl-CoA molecules. Hence, the discovery of FnsA has revealed the mode of naringenin biosynthesis distinct from known plant pathways.

For the formation of naringenin catalysed by FnsA, its A-T module catalyses two inseparable "half" reactions, i.e., adenylation and thioesterification[39]. The FnsA A domain replaces the function of 4CL and is responsible for the recognition and activation of *p*-CA, to form *p*-coumaroyl-AMP. Generally, A domains act as gatekeepers for substrate selection and activation with typically high selectivity. The substrates for NRPS assembly lines derive from proteinogenic and non-proteinogenic amino acids and other aryl acids with carboxylic group[39]. However, it has been reported that A domains can also exhibit a significant substrate promiscuity[40,41]. Our results showed that the A domain of FnsA can adenylate both *p*-CA and *p*-HBA (Figs. 3 and 4), indicating that the A domain has potential for broader substrate tolerance in comparison to the strict substrate selectivity of 4CL[42]. In addition, only acyl-O-AMP was detected in the enzymatic products of FnsA, which may be due to the fact that the protein structure of the

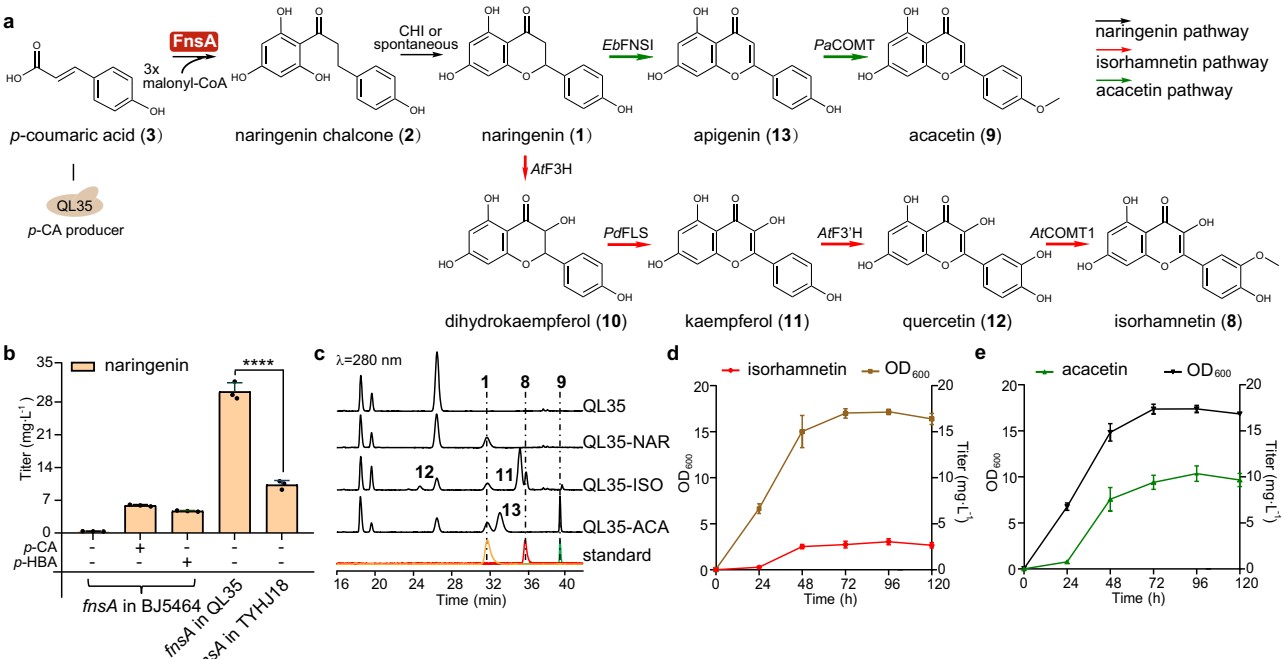

**Fig. 6 | De novo biosynthesis of plant flavonoids in *S. cerevisiae*. a** Schematic illustration of the engineered biosynthetic pathways leading to the production of plant flavonoids isorhamnetin (**8**) and acacetin (**9**). *Eb*FNSI, flavone synthase I from *E. breviscapus*; *Pa*COMT, flavone 4′-O-methyltransferase from *P. appendiculatum*; *At*F3H, flavanone 3-hydroxylase from *A. thaliana*; *Pd*FLS, flavonol synthase from *P. deltoides*; *At*F3′H, flavonol 3′-hydroxylase from *A. thaliana*; *At*COMT1, O-methyltransferase 1 from *A. thaliana*. **b** Yield assessment of naringenin (**1**) supplying with or without the substrate *p*-CA or *p*-HBA in different engineered yeast strains. BJ5464, *S. cerevisiae* BJ5464-NpgA; QL35, *p*-CA high production strain; TYHJ18, *p*-HBA production strain. Cultures were collected after 96 h of growth for metabolite detection. **c** HPLC analysis of the products from QL35-NAR, QL35-ISO and QL35-ACA. The HPLC chromatograms of standards **1**, **8**, and **9** are presented in yellow, red, and green, respectively. **d** Production profile of isorhamnetin during the cultivation in shake flasks. **e** Production profile of acacetin during the cultivation in shake flasks. Statistical analysis was performed by using *t* test (two-tailed; ****$p < 0.0001$). All data represent the mean of *n* = 3 biologically independent samples and error bars show standard deviation (**b, d, e**). Source data are provided as a Source Data file.

PKS portion of FnsA is unstable in vitro. Optimization of the protein purification conditions will help stabilize the protein structure of FnsA. Interestingly, phylogenetic analysis of the KS domain of FnsA revealed that it is close to that of type I PKSs (Fig. 5a), which use substrates tethered to phosphopantetheinyl groups, in contrast to the traditional type III PKSs that require CoA-activated substrates for biosynthesis[43]. This suggested that FnsA belongs to an unidentified type of naringenin synthase (Fig. 5b).

Isorhamnetin is a natural flavonoid, mostly found in the medicinal plant sea-buckthorn and is reported as a potential therapeutic candidate against COVID-19[44]. The bioactive flavonoid acacetin possesses neuroprotection, antinociception and antidepressant-like activity[45]. Based on the catalytic advantages of FnsA's one-protein synthesis of naringenin, we achieved the de novo biosynthesis of naringenin, isorhamnetin and acacetin, with *fnsA*. The successful engineering of those pathways led to product yields of 30.2, 3.1, and 10.4 mg L⁻¹ for naringenin, isorhamnetin and acacetin, respectively (Fig. 6). Although the yield of naringenin is relatively low in comparison with 1,184 mg L⁻¹ from optimized *S. cerevisiae* strain by expression of 4CL, CHS, and CHI[46], the refactoring isorhamnetin biosynthetic pathway has not been reported prior to this study. Furthermore, according to Wang et al.[33], the acacetin product yield achieved to 20.3 mg L⁻¹ under co-cultivation of three engineered strains, but merely up to 2.7 mg L⁻¹ under mono-cultivation[47]. In our case, the titer of acacetin attained 10.4 mg L⁻¹ under our cultural condition. These results provide a promising strategy to achieve natural/non-natural flavonoid synthesis. The product yields could be improved by different strategies such as increasing the intracellular supply of malonyl-CoA[48]. In addition, the codon optimization of the *fnsA* sequence may improve the expression level of FnsA.

In conclusion, we discovered an unexplored naringenin synthase, FnsA, which can catalyse naringenin production using *p*-CA or *p*-HBA as substrate. As a proof of concept, we applied *fnsA* to the de novo biosynthesis of plant flavonoids isorhamnetin and acacetin with product yields of up to 3.1 and 10.4 mg L⁻¹, respectively. This work expands the repertoire of naringenin biosynthesis in nature and provides a strategy for discovery of pathways of unknown natural products.

## Methods

### Strains and culture conditions

The strains used in this study are listed in Supplementary Data 2. *P. fici* CGMCC3.15140 and its transformants were grown at 25 °C on potato dextrose agar (PDA) or potato dextrose broth (PDB) with appropriate antibiotics as required. *Aspergillus nidulans* LO8030[24] and its transformants were grown at 37 °C on glucose minimum medium (GMM) [1.0% (w/v) glucose, salt solution (50 mL L⁻¹), trace element solution (1 mL·L⁻¹) and 1.6% (w/v) agar] with appropriate supplements corresponding to the auxotrophic markers[49]. The salt solution comprises (w/v) 12% NaNO₃, 1.04% KCl, 1.04% MgSO₄ • 7H₂O, and 3.04% KH₂PO₄. The trace element solution contains (w/v) 2.2% ZnSO₄ • 7H₂O, 1.1% H₃BO₃, 0.5% MnCl₂ • 4H₂O, 0.16% FeSO₄ • 7H₂O, 0.16% CoCl₂ • 5H₂O, 0.16% CuSO₄ • 5H₂O, 0.11% (NH₄)₆Mo₇O₂₄ • 4H₂O, and 5% Na₄EDTA. *Escherichia coli* DH5α and BAP1 were cultivated in LB medium with appropriate antibiotics. *Saccharomyces cerevisiae* BJ5464-NpgA[26] and QL35[29] were grown at 28 or 30 °C on synthetic dextrose complete (SDC) medium with appropriate supplements corresponding to the auxotrophic markers[47], YPD with appropriate antibiotics as required or fresh minimal medium (MM)[29]. The *URA3* marker was removed and selected against on SDC with 0.8 g L⁻¹ 5-fluoroorotic acid (SDC + 5-FOA) plates.

## Bioinformatics analysis of NRPS-PKS hybrids and phylogenetic tree construction

502 sequences of NRPS-PKS hybrid enzymes were downloaded from NCBI database. Multiple sequence alignments were performed using MUSCLE. The phylogenetic trees were constructed using neighbour-joining method with bootstrap of 1000 in MEGA6 and the trees were visualized by using the Interactive Tree of Life (ITOL, http://itol.embl.de/). The amino acid sequence of FnsA was used as a query to conduct BLASTp analysis. Domain structures were analysed on the PKS/NRPS website (at http://nrps.igs.umaryland.edu).

## Gene cluster and gene sequence analysis

The biosynthetic gene clusters of *P. fici* were analysed by using anti-SMASH (http://antismash.secondarymetabolites.org/). Prediction of the open reading frames (ORFs) was performed with the online BLAST approaches (http://blast.ncbi.nlm.nih.gov). The gene boundary between the NRPS and PKS portions of FnsA was predicted by NCBI Basic Local Alignment Search Tool search for protein (BLASTp), anti-SMASH, PKS/NRPS and NRPS-PKS analysis tool (http://www.nii.ac.in/~zeeshan/search_only_nrps.html).

## Gene cloning and plasmid construction

The plasmids used for gene deletion and expression in this study are listed in Supplementary Data 2. Genomic DNA (gDNA) of *P. fici* was isolated from mycelia grown in 3 mL PDB at 25 °C for 24 h. Total RNAs from the mycelia of *P. fici* were extracted by using the TranZol™ kit (TransGen Biotech, China) which were cultivated on PDA at 25 °C for 5 days. The same method was used for total RNAs extraction from *Arabidopsis thaliana*. The single-strand cDNA was synthesized by the Fast Quant RT Kit (Tiangen Biotech, China) according to the standard manufacturer's instruction. TransStart ® FastPfu DNA polymerase (TransGen Biotech, China) was used to perform PCR reactions from gDNA or cDNA. Complementary overhangs of 30–35 bp were designed for sequence assembly. Double-joint PCR amplifications were carried out for homologous recombination to create corresponding cassettes[49]. The oligonucleotide sequences for PCR primers are given in Supplementary Data 3.

To construct the deletion cassette of *fnsA*, around 1 kb DNA fragments locating upstream and downstream of the *fnsA* coding region were amplified from the gDNA of *P. fici*. The selection marker gene hygromycin B (*hph*) was amplified from the pYXW15 vector. The three fragments were integrated into the *NdeI/PmeI*-cleaved vector pXW06 via homologous recombination in *S. cerevisiae* BJ5464-NpgA strain to obtain the plasmid pYXW4.

For heterologous expression of *fsnA* in *A. nidulans* LO8030, the coding sequence of *fnsA* gene was amplified from *P. fici* gDNA and combined with *gpdA* promoter as mentioned above to yield plasmid pYWL82 for constructing strain TYWL54. For heterogenous expression of *fnsA*, *fnsA*^A-T, and *fnsA*^PKS in *S. cerevisiae* BJ5464-NpgA (Supplementary Fig. 3 and Supplementary Data 4), the corresponding regions of *fnsA* were amplified from the cDNA of *P. fici* and combined with *ADH2p* promoter as mentioned above to obtain pYZX10, pYHJ10, and pYZS25, respectively. They were transformed into *S. cerevisiae* BJ5464-NpgA to generate strain TYZX19, TYHJ6, TYHJ7 and TYHJ8, respectively. For protein production, the corresponding sequences were fused to a C-terminal *His₆*-tag. For heterologous expression of *fnsA* in QL35, *npgA* amplified from the gDNA of *A. nidulans* LO8030 was inserted, together with *PGK1p* promoter, into pYZX10 as mentioned above to obtain pYHJ13. pYHJ13 was transformed into QL35 to generate QL35-NAR. For heterologous expression of *fnsA*^A-T in *E. coli* BAP1, the *fnsA*^A-T gene was amplified from *P. fici* cDNA and inserted into pET28a to get pYHJ5 through quick-change method[50]. The *fnsA*^A-T sequence was fused to an N-terminal *His₆*-tag for protein purification.

For the construction of integration cassettes in engineered yeast, all gene overexpression cassettes were integrated into designed chromosomal loci that have been demonstrated to provide stable and high-level heterologous gene expression. By taking the example of the targeted integration of *ubiC* and *aroL* at the *TRP1* locus in *S. cerevisiae* BJ5464-NpgA, an around 100 bp DNA fragment upstream of the *TRP1* start codon (*TRP1 us*) and an around 100 bp DNA fragment downstream of the *TRP1* stop codon (*TRP1 ds*) were amplified from the gDNA of *S. cerevisiae* BJ5464-NpgA. *TEF1p*, *ADH1t*, *PGK1p* and *CYC1t* were amplified by PCR using QL35 gDNA as the template. The coding sequences of *ubiC* and *aroL* were amplified from *E. coli* DH5α gDNA. The *Leu2D* marker gene originated from pESC-leu2d. *TRP1 us*, *Leu2D*, and *ADH1t* were jointed to fragment I, *ubiC*, *TEF1p*, and *PGK1p* to fragment II, and *aroL*, *CYC1t* and *TRP1 ds* to fragment III by using the double-joint method. The three fragments were then inserted into *NotI/KpnI*-cleaved vector pXW55 to produce plasmid pYHJ17 by using the Clone Express® MultiS One Step Cloning Kit (Vazyme Biotech Co. Ltd, China). The integration cassette (*TRP1 us-Leu2D-ADH1t-ubiC-TEF1p-PGK1p-aroL-CYC1t-TRP1 ds*) was amplified from pYHJ17 using primer pairs TRP1-F and TRP1-R. Plasmids pYHJ54 and pYHJ55 were constructed in a similar way. The integration cassettes of isorhamnetin (*XII-1 us-TEF1p-AtF3H-TDH2t-TDH3p-PdFLS-CYC1t-TPI1p-AtF3'H-IDP1t-GPM1p-AtCOMT1-FBA1t-XII-1 ds*) and acacetin (*XII-1 us-TEF1p-EbFNSI-TDH2t-TDH3p-PaCOMT-CYC1t-XII-1 ds*) were obtained by digesting pYHJ54 and pYHJ55 with *SbfI* and *AscI*. The genes mentioned above are listed in Supplementary Table 2. The codon-optimized sequences *PdFLS*, *EbFNSI*, and *PaCOMT* (Supplementary Data 5) were synthesized by Tsingke Biological Technology. *XII-1 us* and *XII-1 ds* sequences (Supplementary Data 6) were amplified by PCR using QL35 gDNA as the template.

To construct the gRNA plasmid pYHJ60, a fragment containing the *XII-1* gRNA sequence was amplified using primer pairs pYHJ60-1-F and pYHJ60-1-R and inserted into the vector p426-SNR52p-gRNA.CAN1.Y-SUP4t[29] through quick-change method. To select for specific guide RNAs, all potential gRNAs for the genomic locus *XII-1* were compared with all potential off-targets in the entire CEN. PK113-7D genome by using CRISPRdirect tool (at http://crispr.dbcls.jp/).

For the construction of the deletion cassettes of *ARO7* and *TRP3*, two 100 bp sequences homologous to upstream and downstream regions of the target sites and resistance genes were constructed by double-joint PCR. The selection marker gene *zeocin* was amplified from the vector pPPt4 and the selection marker gene *neo* was amplified from the vector pPIC9K.

## Deletion of *fnsA* in *P. fici*

For the deletion of *fnsA* (*PFICI_04360*), the knock-out cassette of *fnsA* was amplified from pYXW4 using primer pairs KO4360-5F-F and KO4360-3F-R. The knock-out cassette was transformed into *P. fici* by the polyethylene glycol (PEG)-mediated fusion of protoplasts according to the described protocol[47]. Hygromycin B resistant colonies were selected after cultivating on PDA at 25 °C for 5 days. The disruption mutants were verified using diagnostic PCR with primers inside and outside the corresponding gene (Supplementary Fig. 1). PCR screening for transformants was carried out with 2 × High Fidelity PCR Master Mix (Sangon Biotech, China).

## Heterologous expression of *fnsA* in *A. nidulans* LO8030

*A. nidulans* LO8030 was used as the recipient host. Fungal protoplast preparation and transformation were performed by the PEG-mediated fusion of protoplasts according to the described protocol[24]. The plasmid pYWL82 containing *fnsA* and the empty vector pYWL27 were linearized with *SwaI* and transformed into the host strain *A. nidulans* to create the *fnsA* overexpression strain TYWL54 and the control strain TYWL8, respectively. The transformants were selected on GMM with appropriate supplements. All mutants were verified by diagnostic PCR with primers (Supplementary Fig. 2).

## Secondary metabolite analysis

*P. fici* and its transformants were cultivated on PDA at 25 °C for 4 days. The cultures were then cut with a sterile scalpel into 4 mm$^2$ squares and cultivated on 10 g rice medium evenly and grown at 25 °C for 7 days. *A. nidulans* and its transformants were cultivated on GMM medium supplemented with appropriate nutrients at 37 °C for 3 days. The spores were collected with 0.1% Tween-80 and counted in the blood cell counting chamber. $1 \times 10^7$ spores were cultivated on 10 g rice medium at 25 °C for 9 days supplemented with appropriate nutrients. Secondary metabolites were extracted repeatedly with EtOAc and evaporated under reduced pressure. The extracts were dissolved in 1 mL methanol (MeOH) and then analysed on HPLC or LC-MS.

## Feeding experiment in *S. cerevisiae* BJ5464-NpgA

For feeding experiments, the precursors were dissolved in DMSO to give 0.1 M stock solutions. *S. cerevisiae* BJ5464-NpgA strain carrying pYZX10 was cultivated in 3 mL SDC with appropriate supplements at 30 °C for 36 h. The seeds were then grown at 28 °C in 100 mL YPD media with 1% dextrose for 72 h. The cells were harvested by centrifugation (3,214 g, 10 min), resuspended in 10 mL fresh YPD media with 1% dextrose. 0.5 mM precursors were added afterwards to the YPD culture of *S. cerevisiae* BJ5464-NpgA strain carrying pYZX10 for incubation at 30 °C for 48 h. For the time course of the feeding experiments, *S. cerevisiae* BJ5464-NpgA strain carrying pYZX10 was cultivated for 6, 12, 24, and 48 h supplemented with **3** or **4**, respectively. The cultures were extracted repeatedly with EtOAc and the organic solvent was evaporated to dryness, redissolved in MeOH and then analysed on Waters HPLC system. Water with 0.1% formic acid (A) and acetonitrile with 0.1% formic acid (B) was used as the solvent at a flow rate of 1 mL min$^{-1}$. Extracts were eluted with linear gradient from 25 to 43% B in 20 min, 43 to 100% B in 5 min, washed with 100% solvent B for 6 min, and equilibrated with 25% solvent B for 7 min. The feeding experiments of TYHJ6, TYHJ7 and TYHJ8 were conducted using the same method as above.

For determination of the substrate preference of FnsA toward **3** or **4**, the *fnsA*-containing strain was cultivated in YPD media with 1% dextrose at 28 °C for 72 h. The cells were allocated to 50 mL flasks with 10 mL cultures and fed with **3** or **4**, respectively. The concentration of substrate **3** or **4** was 0.01, 0.02, 0.05, 0.1, 0.2, 0.5, and 1 mM. The cultures were incubated at 28 °C for 48 h and extracted twice with one volume EtOAc. The extracts were evaporated and dissolved in 200 μL MeOH and centrifuged at 15,871 g for 20 min before further analysis on HPLC.

## Large-scale fermentation, product purification and structure characterization

To isolate **1** and **2**, the spores of transformant were inoculated into 5 kg rice medium and incubated at 25 °C for 12 days. The rice cultures were extracted with EtOAc by exhaustive maceration, and the organic solvent was concentrated under vacuum to afford a crude extract (10.3 g). The residues were subjected to silica gel column chromatography (CC) and eluted with a gradient of CH$_2$Cl$_2$-MeOH mixtures (100:0-0:100, v/v) to give eight fractions (fractions A-H). Fr. C (55 mg) was subjected to Sephadex LH-20 CC eluting with MeOH to obtain ten subfractions (Fr. 1–10). The target compound **1** (4 mg) was purified from Fr. 6 by semi-preparative HPLC (MeOH-H$_2$O, 60:40). Fr. B (26 mg) was subjected to Sephadex LH-20 CC eluting with MeOH to obtain fifty subfractions (Fr. 1–50). The target compound **2** (0.5 mg) was obtained in Fr. 40–50.

To isolate **5**, *S. cerevisiae* carrying *fnsA* was cultivated in 8 L YPD medium with 1% dextrose in the presence of *p*-CA at a final concentration 0.5 mM. The cultures were extracted with EtOAc, and the organic solvent was evaporated to dryness under vacuum to afford the crude extract (2.7 g). The residues were subjected to silica gel CC using a gradient elution with CH$_2$Cl$_2$-MeOH (100:0-0:100, v/v) to give six fractions (fractions A-F). Fr. B (40 mg) was subjected to Sephadex LH-

20 CC eluting with MeOH to afford ten subfractions (Fr. 1–10). The target compound **5** (2 mg) was purified from Fr. 5-6 by semi-preparative HPLC (MeOH-H$_2$O, 60:40).

Semipreparative purification on HPLC was performed on an SSI HPLC system (Teledyne SSI Lab Alliance Series III pump system and Series 1500 Photodiode Array Detector) with an ODS column (C18, 10.0 by 250 mm, 5 μm, YMC) and a flow rate of 2 mL min$^{-1}$.

NMR spectra ($^1$H, $^{13}$C) were recorded on a Bruker Avance-500 MHz spectrometer using TMS as internal standard (Bruker Corporation, Karlsruhe, Germany). All spectra were processed with MestReNova 12.0 (Metrelab). Chemical shifts are referenced to those of the solvent signals.

(*S*)-Naringenin (**1**), Brown solid. HR-ESI-MS *m/z*: [M + H]$^+$, Calcd for C$_{15}$H$_{13}$O$_5$, 273.0758, found 273.0761. [α]$^{25}_D$ = −19.23° (c = 0.01, MeOH)[25], $^1$H NMR (500 MHz, Methanol-*d*$_4$) δ 7.31 (d, *J* = 8.6 Hz, H-2', H-6', 2H), 6.80 (d, *J* = 8.6 Hz, H-3', H-5', 2H), 5.89 (dd, *J* = 14.0, 2.0 Hz, H-6, H-8, 2H), 5.32 (dd, *J* = 12.6, 3.1 Hz, H-2, 1H), 3.11 (dd, *J* = 17.0, 12.7 Hz, H-3, 1H), 2.70 (dd, *J* = 17.0, 3.1 Hz, H-3, 1H). $^{13}$C NMR (125 MHz, Methanol-*d*$_4$): δ$_C$ 197.8 (C-4), 168.3 (C-7), 165.5 (C-5), 164.9 (C-9), 159.0 (C-4), 131.1 (C-1'), 129.0 (C-2', C-6'), 116.3 (C-3', C-5'), 103.4 (C-10), 97.0 (C-6), 96.1 (C-8), 80.5 (C-2), 44.0 (C-3). The NMR data of **1** correspond well to those of naringenin[51].

Naringenin chalcone (**2**), Yellow solid. HR-ESI-MS *m/z*: [M + H]$^+$, Calcd for C$_{15}$H$_{13}$O$_5$, 273.0758, found 273.0757. $^1$H NMR (500 MHz, Acetone-*d*$_6$): δ$_H$ 8.12 (d, *J* = 15.5 Hz, H-α, 1H), 7.75 (d, *J* = 15.5 Hz, H-β, 1H), 7.56 (d, *J* = 8.6 Hz, H-2, H-6, 2H), 6.91 (d, *J* = 8.6 Hz, H-3, H-5, 2H), 5.97 (s, H-3', H-5', 2H). $^{13}$C NMR (125 MHz, Acetone-*d*$_6$): δ$_C$ 193.2 (C = O), 165.7 (C-4'), 165.5 (C-2', C-6'), 160.6 (C-4), 143.3 (C-α), 131.2 (C-2, C-6), 128.1 (C-1), 125.3 (C-β), 116. 8 (C-3, C-5), 105.7 (C-1'), 96.5 (C-3', C-5'). The NMR data of **2** correspond well to those of naringenin chalcone[52].

4-hydroxystyrene (**5**): White solid. HR-ESI-MS *m/z*: [M + H]$^+$, Calcd for C$_8$H$_9$O, 121.0648, found 121.0649. $^1$H NMR (500 MHz, Methanol-*d*$_4$): δ 7.25 (d, *J* = 8.5, H-3, H-5, 2H), 6.73 (d, *J* = 8.5, H-2, H-6, 2H), 6.62 (dd, 17.6, 10.4, H-7, 1H), 5.55 (d, *J* = 17.6, H-8, 1H), 5.02 (d, *J* = 10.4, H-8, 1H). $^{13}$C NMR (125 MHz, Methanol-*d*$_4$): δ 158.4 (C-1), 137.8 (C-2, C-6), 130.7 (C-4), 128.4 (C-3, C-5), 116.2 (C-8), 110.7 (C-7). The NMR data of **5** correspond well to those of 4-hydroxystyrene[53].

## Expression and purification of FnsA, FnsA$^{PKS}$ and FnsA$^{A-T}$ proteins

For the expressions of FnsA or FnsA$^{PKS}$, *S. cerevisiae* BJ5464-NpgA carrying pYZX10 or pYZS25 was cultivated in SDC with appropriate supplements at 30 °C for 36 h. The cells were grown at 28 °C, 220 rpm in 1 L YPD media with 1% dextrose for 72 h. The cells were harvested by centrifugation at 4 °C and 4629 × *g* for 10 min. The pellets were resuspended in 30 mL lysis buffer (50 mM NaH$_2$PO$_4$, 300 mM NaCl, 10 mM imidazole, pH 8.0) and lysed by liquid nitrogen grinding. Cellular debris was removed by centrifugation at 4 °C and 12,857 g for 10 min. Ni-NTA agarose resin was added to the supernatant (500 μL L$^{-1}$ of culture) and the solution was gently mixed at 4 °C for 2 h. The protein-resin mixtures were loaded into a gravity flow column and eluted with increasing concentration of imidazole (20–40 mM) in buffer (50 mM NaH$_2$PO$_4$, 300 mM NaCl, pH 8.0). The collected protein fraction was desalted by eluting with Tris buffer (50 mM, pH 7.5, 15% glycerol) before being concentrated, aliquoted and flash frozen. Protein concentration was determined by using Nano-Drop (Thermo-Fisher Scientific) to be around 1.0 mg L$^{-1}$. Purity of the protein was confirmed by SDS-PAGE.

To obtain the FnsA$^{A-T}$ protein, *E. coli* BAP1 cells harbouring pYHJ5 were grown in 3 mL LB media with 50 μg mL$^{-1}$ kanamycin at 37 °C for 12 h. The cells were then transferred in 1 L LB media and kept at 37 °C until the optical density (OD$_{600}$) reached 0.6. FnsA$^{A-T}$ expression was induced with 0.1 mM IPTG at 16 °C for 16 h. Cell pellets were harvested by centrifugation at 4 °C and 3214 × *g* for 10 min, and resuspended in 20 mL lysis buffer (50 mM NaH$_2$PO$_4$, 300 mM NaCl, 10 mM

imidazole, pH 8.0). After disrupting the cells by sonication on ice, the mixtures were centrifuged at $12,857 \times g$ and 4 °C for 30 min. The supernatant was mixed with Ni-NTA agarose resin for 2 h at 4 °C. The protein-resin mixtures were loaded into a gravity flow column and FnsA$^{A-T}$ protein was purified with 20 mM imidazole in buffer (50 mM NaH$_2$PO$_4$, 300 mM NaCl, pH 8.0). The collected protein solution was desalted with Tris buffer (50 mM, pH 7.5, 15% glycerol) before being concentrated, aliquoted and flash frozen. Protein concentration was determined on a Nano-Drop C2000 (ThermoFisher Scientific) to be around 15.0 mg L$^{-1}$. Purity of the protein was confirmed by SDS-PAGE.

## In vitro enzyme assay of FnsA

To determine the function of FnsA, the enzyme assays (100 μL) contained Tris-HCl buffer (50 mM, pH 7.5), malonyl CoA (2 mM), ATP (10 mM), MgCl$_2$ (10 mM), DTT (1 mM), **3** and **4** (0.5 mM), and the purified recombinant FnsA (10 μM). The reactions were incubated at 30 °C for 12 h and terminated with equal volume of methanol. The reaction mixtures were centrifuged at $15,871 \times g$ for 30 min to remove the protein precipitate before further analysis on HPLC or LC-MS.

## Load 3 or 4 to FnsA T domain generating modified FnsA T-pantetheine

The enzyme assays (300 μL) contained Tris-HCl buffer (50 mM, pH 7.5), ATP (10 mM), MgCl$_2$ (10 mM), **3** or **4** (0.5 mM), and the purified recombinant FnsA$^{A-T}$ (10 μM). The reactions were incubated at 25 °C for 12 h and loaded on SDS-PAGE afterwards. The target protein band was cut out and digested overnight at 37 °C using 1 μg of proteomics grade trypsin (1:200 w/w) in 50 μL of NH$_4$HCO$_3$ reaction buffer (20 mM, pH 8.2). Peptide fragments were then reduced with 5 mM DTT for 20 min at 50 °C, and thiols were capped with 15 mM iodoacetamide. Following proteolysis, the resulting mixtures of tryptic peptides from each *holo*-FnsA$^{A-T}$ reaction with or without the supplementary of substrate **3** or **4**, were fractionated using a liquid chromatography high-resolution mass spectrometry (LC-HRMS) analysis.

## Hydroxylamine trapping experiment

For adenylation activity assay[21], the reaction (100 μL) contained Tris buffer (25 mM, pH 8.0), ATP (2.25 mM), MgCl$_2$ (15 mM), hydroxylamine (150 mM), tested substrates (3 mM), and the purified recombinant FnsA (2.5 μM). The reactions were incubated at 30 °C for 3 h and terminated with equal volume of stopping solution [10% (w/v) FeCl$_3$ and 3.3% (w/v) trichloroacetic acid (TCA) dissolved in 0.7 M HCl]. The reaction mixtures were centrifuged at $15,871 \times g$ for 30 min. The supernatant was transferred to a 96-well plate, and its absorbance was measured by using a BIO-RAD Imark at 540 nm.

## Construction of *p*-hydroxybenzoic acid-producing yeast strains

Gene expression of *ubiC* and *aroL* in *S. cerevisiae* BJ5464-NpgA was conducted using an integration cassette according to the described protocol[54] and the transformants were selected on SDC-Leu plates as described to obtain TYHJ16. The gDNA of the transformants was extracted and verified by diagnostic PCR using primers listed in Supplementary Data 3.

Gene disruption of *ARO7* in *S. cerevisiae* TYHJ16 was conducted using a knock-out cassette according to the published protocol[54] and the transformants were selected on YPD with Zeocin (200 μg mL$^{-1}$) plates as described to obtain TYHJ17. Gene disruption of *TRP3* in *S. cerevisiae* TYHJ17 was conducted using a knock-out cassette according to the published protocol[54] and the transformants were selected on YPD with G418 (300 μg mL$^{-1}$) plates as described to obtain TYHJ18. The gDNA of the transformants was extracted and verified by diagnostic PCR with primers listed in Supplementary Data 3 (Supplementary Fig. 11).

## Construction of flavonoid-producing yeast strains

QL35 harbours an integrated Cas9 expression cassette under the control of constitutive *TEF1* promoter[29]. For gene expression, DNA integration constructs were integrated at *XII-1* genomic locus[54] via the *CRISPR/Cas9* system. Integration of isorhamnetin or acacetin pathways in QL35 was conducted by using corresponding integration cassettes and gRNA vector pYHJ60, respectively, according to the described protocol[54] and transformants were selected on SDC-URA plates. The gDNA of the transformants was extracted and verified by diagnostic PCR. Subsequently, the *URA3* marker gene of the right transformant was recycled according to the protocol described[29] to obtain TYHJ46 and TYHJ47. To produce isorhamnetin and acacetin, pYHJ13 was transformed into TYHJ48 (QL35-ISO) and TYHJ49 (QL35-ACA), respectively (Supplementary Fig. 13).

## Metabolite extraction and quantification of engineered yeast strains

Three independent single colonies, with the relevant genetic modifications, were inoculated into 5 mL of SDC medium and cultivated overnight at 30 °C and 220 rpm. Next, the seeds were inoculated into 50 mL or 100 mL of YPD medium in 1 L shake flasks at an initial OD$_{600}$ value of 0.05 and cultivated at 30 °C, 250 rpm for 120 h. For the extraction of *p*-hydroxybenzoic acid, 0.5 mL of cell culture was mixed with an equal volume of absolute ethanol (100% v/v), vortexed thoroughly, and centrifuged at $15,871 \times g$ for 20 min before HPLC analysis. For flavonoid product extraction, 1 mL of the culture was extracted with 1 mL of EtOAc for three times. The organic phase was then concentrated under reduced pressure and dissolved in MeOH for HPLC analysis. The statistical analyses were performed using GraphPad Prism (version 8).

## HPLC and LC-MS analysis of secondary metabolites

HPLC analysis was conducted with a Waters HPLC system (Waters e2695, Waters 2998, Photodiode Array Detector) using an XTerra MS C18 column (250 by 4.6 mm, 5 μm, Waters). For the secondary metabolites analysis of transformants, water with 0.1% (v/v) formic acid (A) and acetonitrile (B) was used as the solvent at a flow rate of 1 mL min$^{-1}$. Extracts except for the time course feeding experiments, were eluted with linear gradient from 5 to 100% (v/v) B in 30 min, washed with 100% solvent B for 5 min, and equilibrated with 5% solvent B for 5 min. UV absorption was recorded at 280 nm.

For the secondary metabolites detection of the engineered yeast strain, water with 0.1% (v/v) formic acid (A) and acetonitrile (B) were used as the solvent at a flow rate of 1 mL min$^{-1}$. Extracts were eluted with linear gradient from 10 to 30% (v/v) B in 15 min, 30 to 30% B in 15 min, 30 to 100% B in 10 min, washed with 100% (v/v) solvent B for 5 min, and equilibrated with 10% solvent B for 5 min. UV absorption was recorded at 280 nm.

LC-MS analysis was performed on an Agilent HPLC 1200 series system equipped with a single quadrupole mass selective detector and an Agilent 1100LC MSD model G1946D mass spectrometer by using a Venusil XBP C18 column (3.0 by 50 mm, 3 μm, Bonna-Agela Technologies, China). Water (A) with 0.1% (v/v) formic acid and acetonitrile (B) were used as the solvents at a flow rate of 0.5 mL min$^{-1}$. The substances were eluted with a linear gradient from 5 to 100% (v/v) B in 30 min, then washed with 100% solvent B for 5 min, and equilibrated with 5% solvent B for 10 min. The mass spectrometer was set in electrospray positive ion mode for ionization.

## LC-MS/MS analysis of peptides

For the detection of modified FnsA T-pantetheine, LC-MS/MS analysis was performed on an Agilent Accurate-Mass-QTOF LC-MS 6520 instrument (Agilent Technologies Inc., California, U.S.A.) equipped with a nanoLC-Q Exactive EMR mass spectrometer (ThermoFisher Scientific) and an electrospray ionization (ESI) source by using a RP C18

column (75 μm × 20 cm, 3 μm, ThermoFisher Scientific). Water (A) and acetonitrile (B) both with 0.1% (v/v) formic acid were used as the solvents at a flow rate of 0.5 mL·min⁻¹. The substances were eluted with a linear gradient from 4 to 8% (v/v) B in 8 min, 8 to 22% B in 50 min, 22 to 32% B in 12 min, 32 to 90% B in 7 min and washed with 90% (v/v) solvent B for 7 min. The retrieval and identification of protein were performed by using the SEQUEST HT search engine of Thermo Proteome Discoverer (1.4.0.288) from the database (20210816YY0072-contaminations_m.fasata). The retrieves parameters were set as follows: trypsin digestion, two missing cleavage sites, the mass error of precursor ions less than 10 ppm, the mass of fragment ions <20 mDa, the alkylation of cysteine set as fixed modification, the oxidation of methionine and the specific modification of serine set as variable modification. The filter parameters of retrieve results were as follows: Delta Cn less than 0.1; FDR set as 1%. The filter parameter of peptides was set as follows: peptide confidence as high.

### Reporting summary

Further information on research design is available in the Nature Research Reporting Summary linked to this article.

## Data availability

Data supporting the findings of this work are available within the paper and its Supplementary Information files. A reporting summary for this Article is available as a Supplementary Information file. Source data are provided with this paper.

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

## Acknowledgements

We thank Professor Jens Nielsen (Chalmers University of Technology, Sweden) for kindly sharing the QL35 strain. We thank Professors Yihua Chen, Zhaosheng Kong and Yu Fu from Institute of Microbiology, Chinese Academy of Sciences for kindly sharing BAP1 *E. coli* strain, plant *Arabidopsis thaliana*, and the Liquid nitrogen freezing homogenizer, respectively. We thank Professor Yi Zou (Southwest University, China) for helpful discussion and suggestion of the manuscript. We thank Drs. Jinwei Ren and Wenzhao Wang (Institute of Microbiology, CAS) for NMR and MS data collection. This work was supported by National Key Research and Development Program of China [grant no. 2020YFA0907801]; National Natural Science Foundation of China [grant no. 31861133004]; Deutsche Forschungsgemeinschaft (DFG, German Re-search Foundation) – Li844/11-1; Key Research Program of Frontier Sciences, CAS [grant no. ZDBS-LY-SM016 to W.-B.Y.]; Construction of the Registry and Database of Bioparts for Synthetic Biology, CAS [grant no. ZSYS-016 to W.-B.Y.], and China Postdoctoral Science Foundation [YJ20200309 and 2021M693362].

## Author contributions

W.-B.Y. designed the research and supervised the study. H.Z. performed experiments cultivated fungi, performed fermentation, HPLC analysis and conducted in vivo genetic and in vitro biochemical experiments; Z.L. performed protein purification, HPLC analysis, compound isolation and structural elucidation; S.Z. performed compound purification, assisted in molecular biological experiments; H.R. assisted in compound isolation and structural elucidation; Z.S. assisted in phylogenetic analysis; T.Y. provided the strain resources; W.-B.Y., S.-M.L. and H.Z. wrote the manuscript.

## Competing interests

The authors declare no competing interests.
