## [Peer Review File · Nature Communications]

REVIEWER COMMENTS

Reviewer #1 (Remarks to the Author):

The authors describe the discovery of the fungal enzyme FnsA that catalyses formation of naringenin. Heterologous expression of fnsA in *S. cerevisiae* and feeding experiments established p-coumaric acid and p-hydroxybenzoic acid as precursors that are elongated with malonyl-CoA units. Further engineering of the system to overproduce the respective precursors and addition of tailoring enzymes led to the production of the naringenin derivatives isorhamnetin and acacetin. Since flavonoids are ubiquitous compounds in plants and in our nutrition, it is of high interest that a novel fungal biosynthetic pathway is reported here leading to the same naringenin product but using different enzymes. Activation of the phenylpropanoid proceeds in an adenylation domain, which usually occurs in peptide biosynthesis. The PKS part is related to bacterial type I and not type III as chalcone synthase in plants. Interestingly, the fungal solution seems less elegant because it requires several domains (KS-AT-DH-KR-ACP-TE) to do the same job, where chalcone synthase has a single domain performing a long series of enzymatic steps. Strictly speaking, this type of fungal enzyme has been discovered by another group (Kjaerbolling et al, ref 14) through bioinformatics and correctly attributed to flavonoid biosynthesis by them, but this work firstly demonstrates that the enzyme indeed works as postulated. A novel type of flavonoid biosynthesis is quite spectacular and deserves publication in this journal. However, the quality of the text is unacceptable and requires careful revision of the language. The errors are too numerous to be dealt with in the normal review process. Furthermore, the following issues should be considered: Major issues:

- "NRPS" (see title) stands for "nonribosomal peptide synthetase". Obviously, naringenin is not a peptide. Naming an AT domain that activates an acid (not an amino acid) an "NRPS" which ends up not in a peptide is wrong. Notably, there is a more serious problem behind this than just the nomenclature. The acyl-CoA ligases that supply chalcone synthase in plants with coumaroyl-CoA are close phylogenetic relatives of nonribosomal A domains. What is the phylogenetic tree of the A domain from FnsA? Are they really closest to NRPS A domains?
- It is curious that two different substrates are accepted by FnsA. What are the Michaelis-Menten parameters for 3 and 4? Only saturation kinetics can reveal whether one or the other starting material is preferred.
- How does the new method relying on FnsA compare with established methods for naringenin, isorhamnetin or acacetin biosynthesis? Several studies have reported heterologous production of naringenin and others.
- Does FnsAPKS function if it is provided with preactivated CoA esters of p-coumaric acid or p-hydroxybenzoic acid?
- There is a flood of papers reporting pharmacological effects of flavonoids but if high standards are applied, the medical importance of flavonoids should not be overstated (L232: "outstanding potential for drug discovery" ...). Are there any FDA or EMA approved drugs belonging to this class of compounds? Naringenin is mostly infamous for causing toxic effects by interfering with other compounds. Investigating flavonoid biosynthesis is highly interesting and not every natural product has to be a new cure. The importance of flavonoids comes from the facts that humans eat large amounts of them every day and they affect our health, but not in the way typical drugs with nM affinities and specific targets do.

Minor remarks:

- L76-87: It is not clear how the genome mining began. What was the lead the authors were following? The discovery might be easier to understand if the hypothetical chlorflavonin cluster comes first.
- L233-238: Please indicate how the references are relevant for this manuscript or delete the section. This manuscript reports neither a "novel reaction" nor a "novel structure".
- Genes are expressed, not cDNA. Genes can be cloned from cDNA and expressed in yeast, for instance.
- The final paragraph of the introduction is unusually long. It could be shortened to about 50% (L55-73) by removing details.
- L29: "isorhamnetin and acacetin" drops out of the blue. It must be specified with a few words what these compounds are.
- L253: Either the A-domain is a "gatekeeper" or it's "promiscuous", not both.
- L256: Acyl-O-AMP is relevant here, but no aminoacyl-O-AMP.

- All graphs need y-axis labels. That includes HPLC traces. Y-axis labels don't belong into the figure caption ("UV absorption at 280 is illustrated") but into the graph itself.
- Fig. 1a: What are the highlighted enzymes (TAS1, SwnK ...)?
- Fig. 1c should be a main text figure as it fails to provide better understanding than what is mentioned in the text already.
- Fig. 1d and Fig. 2a: The respective peaks in the *fnsA* chromatogram are extremely difficult to see
- Fig 2a/b: Why are two traces exactly repeated?
- Fig. 2c: Maybe use colors to highlight the fate of the precursor in the final molecule?
- Fig. 3c should better be placed in the SI and instead Extended data Fig. 5b would fit better. Why show the AT domains twice?

Reviewer #2 (Remarks to the Author):

Hongjiao Zhang et al.

Title: A fungal NRPS-PKS enzyme catalyzes flavonoid formation

The manuscript by Zhang et. al. describes the first evidence of a NRPS-PKS enzyme that produces narigenin, the key step in flavonoid biosynthesis. Through genome mining of *Pestalotiopsis fici*, the authors identify a sole gene, *fnsA* that encodes a hybrid NRPS-PKS enzyme. Based on a previous hypothesis, the authors believe that this enzyme might be responsible for flavonoid production in yeast. Through gene knockout mutants and heterologous expression, they show that *FnsA* is responsible for narigenin and flavonoid production. By introducing several genes, the authors also show that two industrial relevant flavonoids can be produced in *S. cerevisiae*. Overall, this study improves our knowledge of flavonoid biosynthesis and its origin. Flavonoids are an important pharmaceutical compound with many health benefits.

Generally, the paper has a good flow. The aim of the study is very clear and the arguments for each experiment and results presented follow logical thinking. However, one concern is the use of abbreviations and chemical name shortenings. This makes the paper hard to read and it takes away the focus from the results.

The results presented in this paper are of interest and exciting for the natural product biosynthesis community and deepen the understanding of narigenin biosynthesis in organisms other than plants.

Minor comments

Ln 66: The abbreviation P-HBA has not been introduced

Ln 229: CHS has not been identified in bacteria. Bacteria has type III PKSs, but not a designated CHS. So far, that is exclusive to plants – with *FnsA* being the only exception.

Fig 1A: Remove the methods on how you made the phylogenetic tree. Move it to the method section rather than having it in the figure legend.

Fig 1D: It's difficult to see peaks corresponding to 1 and 2 in the *fnsA* chromatogram. Might be good to zoom in on those peaks.

Ln 288: *fnsA* is not a gene cluster. It is only one gene.

Ln 291: There's a square in front of *fnsA*-mutant

Fig. 2B: The standard of 3 is overlapping the chromatogram of *fnsA*. It does disturb the figure.

Fig 2D: *S. cerevisiae* shows production of 5 without having *fnsA*. Why is that?

Fig 4A: The light blue clade in the phylogenetic tree is labelled as Fungal type III PKS in the paper, but as Fungal type I PKS in the supplementary figure. Double check.

Reviewer #3 (Remarks to the Author):

This is an interesting manuscript in which the authors identify and characterise a novel fungal NRPS-PKS enzyme, FnsA, which catalyses the biosynthesis of naringenin chalcone. In contrast to the type III PKS, CHS, catalysing the formation of naringenin chalcone from coumaroyl-CoA in bacteria and plants, this new type of naringenin chalcone synthase utilises coumaric acid or hydroxybenzoic acid as substrates. Subsequently, the authors characterise the function of FnsA by in vitro and in vivo analysis and employ the enzyme to produce isorhamnetin and acacetin in recombinant yeast. This study provides insight into flavonoid biosynthesis in fungi; however, in contrast to what has been claimed by the authors, fungal 4CL and CHS have previously been identified and characterised. Nonetheless, the novel enzyme offers an alternative pathway to naringenin production, but the low titres that were obtained in yeast, as well as the enzyme's size, which might pose challenges in heterologous expression, raise doubts about whether the new enzyme will be adopted by the metabolic engineering community, especially as 4CL and CHS are already widely characterised and established enzymes in the field.

Detailed comments:

Line 1: Replace "flavonoid" by "naringenin" in the title to be less ambiguous.

Line 19: The authors claim that biosynthesis of naringenin is only reported in plants and bacteria. However, in line 52 they mention that naringenin is also produced by fungi. Please revise the abstract accordingly.

Line 27: Indicate which acids are activated to be more specific.

Line 29: How would the introduction of the isorhamnetin and acacetin biosynthesis pathways provide a shortcut to produce flavonoids? These biosynthesis pathways have been reported previously. The authors may want to be more specific about what is meant by this.

Line 53: The authors claim that the biosynthesis of flavonoids in fungi remains unknown. However, CHS and 4CL have also been found in the fungus *Alternaria* sp. MG1 (<https://doi.org/10.1016/j.foodchem.2020.128972>). Please revise the text accordingly where appropriate.

Line 55: It sounds as if the authors specifically set out to identify the flavonoid biosynthesis pathway in fungi. Has there been any indication before heterologously expressing *fnsA* that it would be involved in flavonoid biosynthesis? If that was the case, please elaborate on this. If the finding was rather accidental (as it sounds according to lines 76 and following) the text needs to be revised where appropriate.

Line 66: Should it be FnsA instead of *fnsA*, as you are talking about the enzyme?

Line 78: Where do the 99% come from? How many NRPS-PKS hybrid enzymes have been reported? Can you add a reference?

Line 89: Would this approach not extract the whole metabolome and not just secondary metabolites? The text may need to be revised.

Line 98: Can you please provide the ¹³C NMR spectrum of compound 1?

Line 106: It might be worth mentioning the role of NpgA, as it is also cloned later to produce flavonoids.

Fig. 2a: You may want to move the dotted line, indicating naringenin, slightly downwards as it is unclear whether the peak in the *fns* HPLC chromatogram is a peak or part of the dotted line.

Line 119: Move the reference to Extended Data Figure 2 here (rather than line 123).

Line 128: Has a different HPLC method been used to produce the data in Figure 1 and Extended Data Figure 3? According to Figure 1, naringenin chalcone elutes before naringenin; in Extended Data Figure 3 it is the other way around. The retention times seem to be exactly the same, just inverted. How do you explain this? Is this a labelling mistake?

Line 226: Please add a reference to support the claim that naringenin has attracted attention for decades.

Line 227: Please add a reference for improved naringenin production by optimising pathway genes.

Line 228: Naringenin has not been produced in reference 25. Please remove the part on precursor supply or find an alternative reference.

Line 228 and following: CHS has also been found in the fungus *Alternaria* sp. MG1 (see earlier comment). Please revise the text accordingly.

Line 253-254: Is this a general statement or referring to the NRPS module of FnsA? If it is a general statement, please add a reference.

Line 256: Please add a reference for the substrate selectivity of 4CL.

Line 270 and following: The naringenin titres obtained by FnsA are fairly low compared to what has

been previously achieved by heterologous expression of 4CL, CHS and CHI. Please add a few references to give more context to what has been previously obtained. Are there any advantages of FnsA over 4CL and CHS?

Line 291: There might be a "Δ" missing before "fnsA-mutant".

Line 303: When were the samples taken? Presumably after 48 h as naringenin chalcone cannot be detected – is that correct?

RESPONSE TO REVIEWERS' COMMENTS

Reviewer 1:

Comments:

The authors describe the discovery of the fungal enzyme FnsA that catalyses formation of naringenin. Heterologous expression of *fnsA* in *S. cerevisiae* and feeding experiments established *p*-coumaric acid and *p*-hydroxybenzoic acid as precursors that are elongated with malonyl-CoA units. Further engineering of the system to overproduce the respective precursors and addition of tailoring enzymes led to the production of the naringenin derivatives isorhamnetin and acacetin.

Since flavonoids are ubiquitous compounds in plants and in our nutrition, it is of high interest that a novel fungal biosynthetic pathway is reported here leading to the same naringenin product but using different enzymes. Activation of the phenylpropanoid proceeds in an adenylation domain, which usually occurs in peptide biosynthesis. The PKS part is related to bacterial type I and not type III as chalcone synthase in plants. Interestingly, the fungal solution seems less elegant because it requires several domains (KS-AT-DH-KR-ACP-TE) to do the same job, where chalcone synthase has a single domain performing a long series of enzymatic steps. Strictly speaking, this type of fungal enzyme has been discovered by another group (Kjaerbølling et al, ref 14) through bioinformatics and correctly attributed to flavonoid biosynthesis by them, but this work firstly demonstrates that the enzyme indeed works as postulated. A novel type of flavonoid biosynthesis is quite spectacular and deserves publication in this journal.

Thank you very much for the positive comment of our manuscript. We appreciate for Kjaerbølling I et al's excellent work and cited it in our manuscript. In their paper, the biosynthetic pathway of chlorflavonin is still exciting and

awaiting characterization. According to your suggestion, we reorganized the result description and mentioned the hypothetical chlorflavonin synthetase at the beginning of the introduction.

However, the quality of the text is unacceptable and requires careful revision of the language. The errors are too numerous to be dealt with in the normal review process. Furthermore, the following issues should be considered:

Thank you very much for the comments. We improved the text's quality with careful language revision and double-checked the mistakes in the manuscript. And the point-by-point responses are stated below.

Major issues:

1. “NRPS” (see title) stands for “nonribosomal peptide synthetase”. Obviously, naringenin is not a peptide. Naming an AT domain that activates an acid (not an amino acid) an “NRPS” which ends up not in a peptide is wrong.

Thank you very much for your comment. I agree with you that an “NRPS” ends up in a peptide normally. However, NRPS-PKS or PKS-NRPS hybrid enzymes are used in the community for proteins with incomplete NRPS domain structures, often only A-T in the “NRPS” part. The products of these enzymes are usually no peptides, e.g. hispidin formation by a NRPS-PKS hybrid HispS (Proc. Natl. Acad. Sci. U. S. A 115 (2018) 12728-32), AnATPPS in the biosynthesis of pyrophen (J. Nat. Prod. 83 (2020) 593-600) and swainsonine biosynthesis by the NRPS-PKS hybrid SwnK (ACS. Chem. Biol. 15(2020), 2476-2484). Our case is an additional example. To avoid confusing, we would like to keep the term “NRPS-PKS” for our enzyme. Of course, we will be happy, if you have a better suggestion.

Notably, there is a more serious problem behind this than just the nomenclature. The acyl-CoA ligases that supply chalcone synthase in plants with coumaroyl-CoA are close phylogenetic relatives of nonribosomal A domains. What is the phylogenetic tree of the A domain from FnsA? Are they really closest to NRPS A domains?

We constructed a phylogenetic tree of A domain from FnsA with known with proteins catalyzing similar reactions. Our data suggest that FnsA A domain is clustered separately from 4CL in plants, bacteria, and fungi. It is grouped with the A domain of NRPS-PKS hybrids from fungi and bacteria. (Please see Figure below). It should be mentioned that these results are based on amino acid sequences. This suggests that FnsA may act as a unique function in the biosynthesis of naringenin.

Phylogenetic analysis of the FnsA A domain.

2. It is curious that two different substrates are accepted by FnsA. What are the Michaelis-Menten parameters for **3** and **4**? Only saturation kinetics can reveal whether one or the other starting material is preferred.

Thank you for your suggestion. We tried to carry out *in vitro* assay with protein FnsA from yeast in the presence of **3** or **4**, but no product was detected, so that no Michaelis-Menten parameters can be obtained. It seems that the recombinant protein doesn't work. Therefore, for determination of substrate preference of FnsA toward **3** and **4**, feeding experiments were performed in the *fnsA*-containing yeast strain. Substrates **3** and **4** with different concentrations of 0.01, 0.02, 0.05, 0.1, 0.2, 0.5, 1 mM were used. Substrate consumption and product yields were calculated. Our results indicated that *p*-CA is likely preferred by FnsA (new figure Supplementary Fig. 9), which is consistent with FnsAA domain specificity assay (Supplementary Fig. 8).

Supplementary Fig. 9 Comparison of product yields toward 3 and 4 by feeding experiments

3. How does the new method relying on FnsA compare with established methods for naringenin, isorhamnetin or acacetin biosynthesis? Several studies have reported heterologous production of naringenin and others.

The engineering strains for naringenin and acacetin production was reported previously by the construction of the biosynthetic pathway requiring at least two enzymes 4CL and CHS. However, no engineering strain for the production of isorhamnetin was reported until our research. Additionally, in our case, the fungal hybrid enzyme FnsA can produce naringenin alone, which reduces the enzyme catalytic steps. Furthermore, the reported approaches can only take p-CA as a substrate for the formation of naringenin, while FnsA accepts both p-CA and p-hydroxybenzoic acid (p-HBA) as substrates. Therefore, our data provide an alternative approach for the production of naringenin and its derivatives. We also add the information in the discussion.

4. Does FnsA PKS function if it is provided with preactivated CoA esters of *p*-coumaric acid or *p*-hydroxybenzoic acid?

Thank you for your suggestion. We performed the *in vitro* assay of FnsA^{PKS} with *p*-coumaroyl-CoA as substrate supplemented with malonyl-CoA. However, the HPLC analysis of reaction extracts showed no formation of naringenin. It is likely that *p*-CA was only accepted by A domain for covalenting with T domain of FnsA. *p*-coumaroyl-CoA can be used neither by A nor by KS domain as substrate. This result is consistent with the phylogenetic analysis of FnsA KS

domain which is far away from type III PKS using *p*-coumaroyl-CoA as substrate.

Supplementary Fig. 7. *In vitro* enzymatic assay of FnsA^{PKS}

a SDS-PAGE analysis of the purified FnsA^{PKS}. FnsA^{PKS} was expressed in *S. cerevisiae* and purified with His₆-tag. The protein yield was 1 mg·L⁻¹. M, marker. **b** HPLC analysis of the enzyme reactions of FnsA^{PKS} with *p*-coumaroyl-CoA. **c** Proposed reactions of FnsA with *p*-CA as substrate and FnsA^{PKS} with *p*-coumaroyl-CoA as substrate.

5. There is a flood of papers reporting pharmacological effects of flavonoids but if high standards are applied, the medical importance of flavonoids should not be overstated (L232: “outstanding potential for drug discovery” ...). Are there any FDA or EMA approved drugs belonging to this class of compounds? Naringenin is mostly infamous for causing toxic effects by interfering with other compounds. Investigating flavonoid biosynthesis is highly interesting and not every natural product has to be a new cure. The importance of flavonoids comes from the facts that humans eat large amounts of them every day and they affect our health, but not in the way typical drugs with nM affinities and specific targets do.

Thank you for the very useful comments. To avoid confusion, we have rephrased some sentences as your suggestions. Indeed, no FDA and EMA approved drugs belong to flavonoids. Compounds from this family are considered valuable nutraceutical ingredients, we removed the highlights of their medical importance.

Minor remarks:

1. L76-87: It is not clear how the genome mining began. What was the lead the authors were following? The discovery might be easier to understand if the

hypothetical chlorflavonin cluster comes first.

Thank you very much for your suggestion. According to your comments, we start the description of FnsA discovery from the hypothetical chlorflavonin cluster. “For example, a NRPS-PKS hybrid (P168DRAFT_323099) from *Aspergillus campestris* was presumed to be involved in the formation of a flavonoid chlorflavonin¹⁴. Thus, we focused on an abundant secondary metabolite fungus *P. fici* and identified a NRPS-PKS hybrid enzyme PFICI_04360 composed of A-T-KS-AT-DH-KR-ACP-TE (A, adenylation; T, thiolation; AT, acyltransferase; DH, dehydratase; KR, ketoacyl reductase; ACP, acyl carrier protein; TE, thioesterase domains) (Fig. 1b), sharing 64.3% amino acid identity with P168DRAFT_323099. However, there is no experimental evidence that this NRPS-PKS or their homologs produce flavonoids in the current cognitive logic.”

2. L233-238: Please indicate how the references are relevant for this manuscript or delete the section. This manuscript reports neither a “novel reaction” nor a “novel structure”.

Many thanks for your comments. We revised these to “genome mining of some special enzymes has been proven to catalyze unique reactions and discover specific structures.” and pointed out the type of indicated enzymes to reveal the unpredictability of the catalytic functions.

3. Genes are expressed, not cDNA. Genes can be cloned from cDNA and expressed in yeast, for instance.

Thanks. We revised the sentence as “we cloned *fnsA* gene from the complementary DNA (cDNA) and expressed it under the control of *ADH2p* promoter in *Saccharomyces cerevisiae* BJ5464-NpgA,”

4. The final paragraph of the introduction is unusually long. It could be shortened to about 50% (L55-73) by removing details.

We have shortened that paragraph as your suggestion.

5. L29: “isorhamnetin and acacetin” drops out of the blue. It must be specified with a few words what these compounds are.

Done. We revised as “refactoring of bioactive flavonoids isorhamnetin and acacetin”

6. L253: Either the A-domain is a “gatekeeper” or it’s “promiscuous”, not both.

We revised the text as “Generally, A domain acts as the gatekeeper for substrate selection and activation with typically highly catalytic selectivity, the substrates for NRPS assembly line derive from proteinogenic and nonproteinogenic amino acids and other aryl acids with carboxylic group³⁷ However, it has been reported that A domain could also present a broad substrate promiscuity in consistent with high activation efficiency³⁸.”

7. L256: Acyl-O-AMP is relevant here, but no aminoacyl-O-AMP.

Thanks. Changed.

8. All graphs need y-axis labels. That includes HPLC traces. Y-axis labels don’t belong into the figure caption (“UV absorption at 280 is illustrated”) but into the graph itself.

Done.

9. Fig. 1a: What are the highlighted enzymes (TAS1, SwnK ...)?

We provided the information on those enzymes in the main text (L73-76) where Fig. 1a was cited.

10. Fig. 1c should be a main text figure as it fails to provide better understanding than what is mentioned in the text already.

We supplied the captions for Fig. 1c as “Deletion of *fnsA* in *P. fici*” and Fig. 1d

as “Heterologous expression *fnsA* in *A. nidulans*” for better understanding.

11. Fig. 1d and Fig. 2a: The respective peaks in the *fnsA* chromatogram are extremely difficult to see.

We exhibited an enlarged chromatogram of product **1** in both Fig. 1d and Fig. 2a.

12. Fig 2a/b: Why are two traces exactly repeated?

We removed the repeated extract chromatogram of the transformant containing *fnsA* gene in Fig. 2b.

13. Fig. 2c: Maybe use colors to highlight the fate of the precursor in the final molecule?

Excellent suggestion. We have done it.

14. Fig. 3c should better be placed in the SI and instead Extended data Fig. 5b would fit better. Why show the AT domains twice?

We made a new figure 3 as your suggestion. In addition, we also made a new figure 4 for a better understanding of A domain function and deleted the repeating AT domain.

Reviewer #2

The manuscript by Zhang et. al. describes the first evidence of a NRPS-PKS enzyme that produces naringenin, the key step in flavonoid biosynthesis. Through genome mining of *Pestalotiopsis fici*, the authors identify a sole gene, *fnsA* that encodes a hybrid NRPS-PKS enzyme. Based on a previous hypothesis, the authors believe that this enzyme might be responsible for flavonoid production in yeast. Through gene knockout mutants and

heterologous expression, they show that FnsA is responsible for naringenin and flavonoid production. By introducing several genes, the authors also show that two industrial relevant flavonoids can be produced in *S. cerevisiae*. Overall, this study improves our knowledge of flavonoid biosynthesis and its origin. Flavonoids are an important pharmaceutical compound with many health benefits.

Generally, the paper has a good flow. The aim of the study is very clear and the arguments for each experiment and results presented follow logical thinking. However, one concern is the use of abbreviations and chemical name shortenings. This makes the paper hard to read and it takes away the focus from the results. The results presented in this paper are of interest and exciting for the natural product biosynthesis community and deepen the understanding of naringenin biosynthesis in organisms other than plants.

Thank you for the positive comments and very useful suggestions. We normalized the abbreviations and chemical name shortenings in the manuscript. And the point-by-point responses are stated below.

Minor comments

1. Ln 66: The abbreviation *p*-HBA has not been introduced.

Done.

2. Ln 229: CHS has not been identified in bacteria. Bacteria has type III PKSs, but not a designated CHS. So far, that is exclusive to plants – with FnsA being the only exception.

Thank you for your comment. We rephrased the sentence in Ln 234-235 as “It is well known that naringenin synthesis requires two key enzymes, 4CL and CHS, to form the skeleton, which are essential module pathways for flavonoids.”

3. Fig 1A: Remove the methods on how you made the phylogenetic tree. Move it to the method section rather than having it in the figure legend.

Done.

4. Fig 1D: It's difficult to see peaks corresponding to 1 and 2 in the *fnsA* chromatogram. Might be good to zoom in on those peaks.

Thanks. We have removed the dotted line covering product peak, and exhibited an enlarged chromatogram of product 1 in both Fig. 1d and Fig. 2a.

5. Ln 288: *fnsA* is not a gene cluster. It is only one gene.

Thanks for your comment. We revised the legend of Fig. 1b as "Portion of the putative *fnsA*-containing gene cluster from *P. fici*."

6. Ln 291: There's a square in front of *fnsA*-mutant

Thanks. We removed the square and added the "Δ".

7. Fig. 2B: The standard of 3 is overlapping the chromatogram of *fnsA*. It does disturb the figure.

Thank you for your suggestion. We removed the repeated extract chromatogram of the transformant containing *fnsA* gene in Fig. 2b.

8. Fig 2D: *S. cerevisiae* shows production of **5** without having *fnsA*. Why is that?

5 was isolated and identified as 4-hydroxystyrene. **3** could spontaneously decarboxylate to **5** by phenylacrylic acid decarboxylase (PAD1) and ferulic acid decarboxylase (FDC1) in *S. cerevisiae* (DOI: 10.1016/j.jbiosc.2009.11.011). Therefore **5-*d*₆** was detected under the feeding of **3-*d*₆** without having *fnsA* in Fig. 2D. Please see Ln 122-126 and Supplementary Fig. 5 for details.

9. Fig 4A: The light blue clade in the phylogenetic tree is labelled as Fungal type III PKS in the paper, but as Fungal type I PKS in the supplementary figure. Double check.

Thank you for your comment. We corrected the annotation as “Fungal type I PKS” for the light blue clade in the revised Fig. 5A.

Reviewer #3

This is an interesting manuscript in which the authors identify and characterise a novel fungal NRPS-PKS enzyme, FnsA, which catalyses the biosynthesis of naringenin chalcone. In contrast to the type III PKS, CHS, catalysing the formation of naringenin chalcone from coumaroyl-CoA in bacteria and plants, this new type of naringenin chalcone synthase utilises coumaric acid or hydroxybenzoic acid as substrates. Subsequently, the authors characterise the function of FnsA by in vitro and in vivo analysis and employ the enzyme to produce isorhamnetin and acacetin in recombinant yeast. This study provides insight into flavonoid biosynthesis in fungi; however, in contrast to what has been claimed by the authors, fungal 4CL and CHS have previously been identified and characterised.

Thank you for the positive comment on our manuscript. We searched for the literature on 4CL and CHS from fungi and changed our statement of no naringenin biosynthetic pathway reported in fungi. The corresponding texts were revised in Lines 52-57 and lines 232-241.

Nonetheless, the novel enzyme offers an alternative pathway to naringenin production, but the low titres that were obtained in yeast, as well as the enzyme's size, which might pose challenges in heterologous expression, raise doubts about whether the new enzyme will be adopted by the metabolic engineering community, especially as 4CL and CHS are already widely characterised and established enzymes in the field.

As your comment, this enzyme offers an alternative way to naringenin

production. For our case, this is just the original level without any engineering. We are doing several ways to improve naringenin production such as optimizing gene expression, precursor malonyl-CoA supply, or discovering an additional homologous enzyme for engineering. Furthermore, the titer of acacetin in our case is 10.4 mg L⁻¹, higher than the reported titer of 4.2 mg L⁻¹ under similar cultivation conditions (DOI: 10.1002/biot.202000131).

The point-by-point responses are stated below.

Detailed comments:

1. Line 1: Replace “flavonoid” by “naringenin” in the title to be less ambiguous.

Thank you very much for your suggestion. We reported the fungal hybrid enzyme FnsA as a naringenin synthase. Furthermore, FnsA was engineered as a key enzyme for the production of plant flavonoids, such as isorhamnetin and acacetin, by the *de novo* biosynthesis approach in the engineered yeast factory. Therefore, the current title will fit the content better.

2. Line 19: The authors claim that biosynthesis of naringenin is only reported in plants and bacteria. However, in line 52 they mention that naringenin is also produced by fungi. Please revise the abstract accordingly.

Many thanks for your comment. To avoid the confusion, we rephrased the texts in Line 19 as “Biosynthesis of naringenin in plants and bacteria is commonly catalyzed by a type III polyketide synthase by using one *p*-coumaroyl-CoA and three malonyl-CoA molecules.”

3. Line 27: Indicate which acids are activated to be more specific.

Thanks. We added “both *p*-CA and *p*-HBA are activated by the adenylation domain of FnsA.”

4. Line 29: How would the introduction of the isorhamnetin and acacetin biosynthesis pathways provide a shortcut to produce flavonoids? These biosynthesis pathways have been reported previously. The authors may want to be more specific about what is meant by this.

Thank you for your suggestion. We revised the abstract as “Ultimately, refactoring of bioactive flavonoids isorhamnetin and acacetin in the engineered *fnsA* yeast provides an alternative approach for the production of flavonoids.”

5. Line 53: The authors claim that the biosynthesis of flavonoids in fungi remains unknown. However, CHS and 4CL have also been found in the fungus *Alternaria* sp. MG1 (<https://doi.org/10.1016/j.foodchem.2020.128972>). Please revise the text accordingly where appropriate.

Thank you very much for your suggestion. We cited this reference and revised the text in Introduction section as “It is worth mentioning that fungi can produce various flavonoids such as naringenin, quercetin, kaempferol, and chlorflavonin^{13,14}. Although 4CL- and CHS-like proteins have been identified in some endophytes^{13,15}, there is no genetic or biochemical evidence for their function.

6. Line 55: It sounds as if the authors specifically set out to identify the flavonoid biosynthesis pathway in fungi. Has there been any indication before heterologously expressing *fnsA* that it would be involved in flavonoid biosynthesis? If that was the case, please elaborate on this. If the finding was rather accidental (as it sounds according to lines 76 and following) the text needs to be revised where appropriate.

Thank you for your comment. A NRPS-PKS hybrid enzyme has been presumed for chlorflavonin biosynthesis by bioinformatics analysis which we also mentioned previously. Therefore, for a better understanding, we revised the text

as “For example, a NRPS-PKS hybrid (P168DRAFT_323099) from *Aspergillus campestris* was presumed to be involved in the formation of a flavonoid chlorflavonin¹⁴. Thus, we focused on an abundant secondary metabolite fungus *P. fici* and identified a NRPS-PKS hybrid enzyme PFICI_04360 composed of A-T-KS-AT-DH-KR-ACP-TE (A, adenylation; T, thiolation; AT, acyltransferase; DH, dehydratase; KR, ketoacyl reductase; ACP, acyl carrier protein; TE, thioesterase domains) (Fig. 1b), sharing 64.3% amino acid identity with P168DRAFT_323099. However, there is no experimental evidence that this NRPS-PKS or their homologs produce flavonoids in the current cognitive logic.”

7. Line 66: Should it be FnsA instead of fnsA, as you are talking about the enzyme?

Thanks. We double-checked and corrected the forms of *fnsA* as a gene and FnsA as a protein through the revised manuscript.

8. Line 78: Where do the 99% come from? How many NRPS-PKS hybrid enzymes have been reported? Can you add a reference?

Genome mining from over 2,000 public fungal genomes led to the identification of 502 fungal NRPS-PKS hybrid enzymes. Among them, only five enzymes (TAS1, SwnK, HispS, AnATPKS, and HppS) were reported for their catalytic functions, suggesting 99% of them are cryptic. Therefore, we cited relating Supplementary Data here and revised the sentence in Lines 74-78 as “Among them, only five enzymes were identified for their catalytic function (TAS1 producing tenuazonic acid¹⁶, SwnK producing swainsonine¹⁷, HispS producing hispidin¹⁸, AnATPKS producing pyrophen¹⁹, and HppS producing 4-hydroxy-6-(4-hydroxyphenyl)- α -pyrone²⁰), indicating that more than 99% enzymes remain to be explored (Fig. 1a and Supplementary Data 1).”

9. Line 89: Would this approach not extract the whole metabolome and not just secondary metabolites? The text may need to be revised.

Thanks for your comments. We revised the text as “High-performance liquid chromatography (HPLC) analysis of metabolite profile from ethyl acetate (EtOAc) extracts of cultures showed no obvious chemical differences between wild type and deletion strains (Fig. 1c)”.

10. Line 98: Can you please provide the ^{13}C NMR spectrum of compound 1?
Done. The ^{13}C NMR spectrum was supplied in Supplementary Figure 17.

11. Line 106: It might be worth mentioning the role of NpgA, as it is also cloned later to produce flavonoids.

Many thanks for your suggestion. We explained the function of NpgA in line 109, as “*Saccharomyces cerevisiae* BJ5464-NpgA, which contains a *holo*-ACP synthase NpgA for PKS- and NRPS-containing gene expression.”

12. Fig. 2a: You may want to move the dotted line, indicating naringenin, slightly downwards as it is unclear whether the peak in the fns HPLC chromatogram is a peak or part of the dotted line.

Thanks. We have removed the dotted line covering product peak, and exhibited an enlarged chromatogram of product 1 in both Fig. 1d and Fig.2a.

13. Line 119: Move the reference to Extended Data Figure 2 here (rather than line 123).

Done.

14. Line 128: Has a different HPLC method been used to produce the data in Figure 1 and Extended Data Figure 3? According to Figure 1, naringenin chalcone elutes before naringenin; in Extended Data Figure 3 it is the other way around. The retention times seem to be exactly the same, just inverted. How do you explain this? Is this a labelling mistake?

Thanks for your comment. This is due to the different analysis methods used for chemical profile detection. We revised the labels and added the details in the Method section.

15. Line 226: Please add a reference to support the claim that naringenin has attracted attention for decades.

We added the reference (ACS. Synth. Biol. 11 (2022) 2339-2347).

16. Line 227: Please add a reference for improved naringenin production by optimising pathway genes.

We added the reference (ACS. Omega. 4 (2019) 12872-12879).

17. Line 228: Naringenin has not been produced in reference 25. Please remove the part on precursor supply or find an alternative reference.

We cited the reference (J. Agric. Food. Chem. 65 (2017) 6638-6646).

18. Line 228 and following: CHS has also been found in the fungus *Alternaria* sp. MG1 (see earlier comment). Please revise the text accordingly.

We cited the reference (Food. Chem. 347 (2021) 128972) and revised the sentence as “It is well known that naringenin synthesis requires two key enzymes, 4CL and CHS, to form the skeleton, which are essential module

pathways for flavonoids.”

19. Line 253-254: Is this a general statement or referring to the NRPS module of FnsA? If it is a general statement, please add a reference.

Many thanks for your comment. Typical substrate specificity of A domain is a general statement, but in some cases, A domain performs occasional promiscuity. Therefore, for better understanding, we revised the text in Lines 255-259 as “Generally, A domain acts as the gatekeeper for substrate selection and activation with typically highly catalytic selectivity, the substrates for NRPS assembly line derive from proteinogenic and nonproteinogenic amino acids and other aryl acids with carboxylic group³⁷ However, it has been reported that A domain could also present a broad substrate promiscuity in consistent with high activation efficiency³⁸.”

20. Line 256: Please add a reference for the substrate selectivity of 4CL.

Done.

21. Line 270 and following: The naringenin titres obtained by FnsA are fairly low compared to what has been previously achieved by heterologous expression of 4CL, CHS and CHI. Please add a few references to give more context to what has been previously obtained. Are there any advantages of FnsA over 4CL and CHS?

Many thanks for your suggestion. This enzyme offers an alternative way to naringenin production. For our case, this is just the original level without any engineering. We are doing several ways to improve naringenin production such as optimizing gene expression, precursor malonyl-CoA supply, or discovering an additional homologous enzyme for engineering. Furthermore, the titer of acacetin in our case is 10.4 mg L⁻¹, higher than the reported titer of 4.2 mg L⁻¹

under similar cultivation condition (DOI: 10.1002/biot.202000131).

In addition, the reported approaches can only take *p*-coumaric acid as substrate for the formation of naringenin, while FnsA accepts both *p*-coumaric acid and *p*-hydroxybenzoic acid as substrates.

Reported:

Our study:

Figure. Biosynthetic pathway of naringenin in previous and our studies.

22. Line 291: There might be a “Δ” missing before “fnsA-mutant”.

Done.

23. Line 303: When were the samples taken? Presumably after 48 h as naringenin chalcone cannot be detected – is that correct?

The samples were analyzed after nine-day cultivation. Therefore, we revised the sentence at line 96 as “the overexpression strain of *PFICL_04360* produced two additional peaks (1 and 2) in HPLC profiles in comparison to that of the control strain after nine-day cultivation (Fig. 1d).”

REVIEWER COMMENTS

Reviewer #1 (Remarks to the Author):

The authors have managed to solve the scientific problems brought up by the reviewer and are congratulated once more for their exciting findings. Unfortunately, the quality of the text remains completely unacceptable for publication in Nature Communications. Due to the large number of problems with the text, the reviewer has made edits in the word document directly (except the Methods section) and submitted it as a PDF. Edits by the reviewer are in purple, problematic sections are highlighted in yellow. Some of these edits are rather drastic. Therefore, the authors are implored to carefully check whether the facts are still correct maybe with the help of online translation tools and to take the text editing process into their own hands from here on.

The following problems remain to be solved:

- Abstract, L16: Are there bacteria naturally making naringenin? Otherwise "bacteria" should be deleted.
- L59: The number of chemical steps for the formation of naringenin is not different from the plant pathway. Only the architecture of the enzyme is different. The fungal enzyme has more domains than the plant enzyme to do basically the same chemical reactions, which rather seems like a disadvantage. Therefore, it is wrong to advertise the fungal enzyme as catalyzing a "one-step reaction". It is also not "a shortcut" (L63).
- L67-83: The search for the gene cluster is still not described in a logical way. The reviewer has made a suggestion, but it is impossible to know for them what really happened. Please check this carefully.
- L83: An orphan gene is normally defined as a gene without homologues in related organisms and has nothing to do with the presence of biosynthetic genes "upstream and downstream". The authors show that there are homologues for the entire cluster in fungi and for individual domains across several domains of life. Why call it an orphan?
- Please add a reference for the substrate promiscuity of A domains (L109).
- L114: Here and in other places, the authors write "3 and 4" or something similar. This could literally mean that both substrates were added at once. If they were added individually, it should be "3 or 4".
- Please note that "covalenting" is not a word. Maybe it should be.
- L169/170: Methionine residues usually oxidize from a thioether to a sulfoxide not to an alcohol.
- For describing the engineered yeast strains (P10ff), it would help to give names to the important strains to be able to precisely refer back to them when another gene is added. For example "QL35-NAR" for the naringenin producer etc.
- L240-244: This part continues to misrepresent what has been discovered in this study. The listed enzymes are examples where novel reaction mechanisms yielding novel structures have been discovered. This is not the novelty of this study. "Special enzyme" is a very blurry description.
- L263: Ref. 38 is not a good reference to support the sometimes promiscuous role of A-domains and there is a typo in the reference title.
- The attached version is a first step towards a readable text. For instance, the grammatical tenses are not ideally or wrongly used throughout the text and have not been carefully corrected in the attached version. More work is needed by a professional editor.
- P11: What is "small fermentation"? Explicit volumes would be helpful.

Reviewer #2 (Remarks to the Author):

Thank you for responding to all the comments I had, and making the necessary changes.

Reviewer #3 (Remarks to the Author):

The revisions made by the authors address most of my concerns and have substantially improved the manuscript. A few issues still remain (numbered according to the response letter), and there are still some unaddressed concerns regarding the novelty of the presented work:

Comment 1, Title: To my understanding FnsA has not been engineered to produce any flavonoid

other than naringenin. The enzyme has been expressed in an engineered strain that allows for the subsequent conversion of naringenin into isorhamnetin and acacetin. Therefore, the title could be amended to "A fungal NRPS-PKS enzyme catalyzes formation of the flavonoid naringenin" to be less ambiguous. The current title still seems to be somewhat overstating the case.

Comment 4, Line 29: The authors may want to revise the sentence. Biosynthetic pathways are refactored, not compounds.

Comment 5, Line 53: The text needs to be revised. CHS and 4CL from *Alternaria* sp. MG1 have been heterologously expressed in *S. cerevisiae*, and their functions have been experimentally validated (<https://pubmed.ncbi.nlm.nih.gov/33453581/>). Furthermore, the text in the discussion (lines 238 and following) needs to be revised accordingly.

Comment 15, Line 226: Please explain how the reference supports the claim that naringenin has attracted attention for decades. The cited paper ("Glycosylation Modification Enhances (2 S)-Naringenin Production in *Saccharomyces cerevisiae*") was published in June 2022 and does not represent decades of attention.

Comment 21, Line 270ff: As requested earlier, in the manuscript please provide more context regarding the concentrations of naringenin that have previously been achieved by expression of 4CL, CHS, and CHI in yeast. It is not sufficient to discuss this in the response to the reviewers. Furthermore, according to Wang et al., acacetin levels of 20.3 mg/L were obtained, which would be substantially higher than the 10.4 mg/L obtained by the authors. Where does the stated comparative titre of 4.2 mg/L come from?

RESPONSE TO REVIEWERS' COMMENTS

Reviewer #1:

Comments:

The authors have managed to solve the scientific problems brought up by the reviewer and are congratulated once more for their exciting findings. Unfortunately, the quality of the text remains completely unacceptable for publication in Nature Communications. Due to the large number of problems with the text, the reviewer has made edits in the word document directly (except the Methods section) and submitted it as a PDF. Edits by the reviewer are in purple, problematic sections are highlighted in yellow. Some of these edits are rather drastic. Therefore, the authors are implored to carefully check whether the facts are still correct maybe with the help of online translation tools and to take the text editing process into their own hands from here on.

Thank you very much for your intensive edits which have been accepted already. In addition, the co-author Prof. Shu-Ming Li and I have done intensive edits for the language problem of the full text including the grammar, tense mistakes etc.

Remarks:

1) Abstract, L16: Are there bacteria naturally making naringenin? Otherwise "bacteria" should be deleted.

Yes. It has been reported that *Streptomyces clavuligerus* produced naringenin naturally by LC-MS analysis and structure elucidation (Álvarez-Álvarez et al. Microb Cell Fact (2015) 14:178). The corresponding reference (Ref. 10) was cited in L40.

2) L59: The number of chemical steps for the formation of naringenin is not different from the plant pathway. Only the architecture of the enzyme is different. The fungal enzyme has more domains than the plant enzyme to do basically

the same chemical reactions, which rather seems like a disadvantage. Therefore, it is wrong to advertise the fungal enzyme as catalyzing a “one-step reaction”. It is also not “a shortcut” (L63).

Thank you for your comment, we have accepted your edits.

3) L67-83: The search for the gene cluster is still not described in a logical way. The reviewer has made a suggestion, but it is impossible to know for them what really happened. Please check this carefully.

We have checked and accepted your edits.

4) L83: An orphan gene is normally defined as a gene without homologues in related organisms and has nothing to do with the presence of biosynthetic genes “upstream and downstream”. The authors show that there are homologues for the entire cluster in fungi and for individual domains across several domains of life. Why call it an orphan?

Thank you very much for your comments. We have carefully checked and revised the relative description. The sentence was revised as “suggesting *PFICL_04360* as an independently functional gene.”

5) Please add a reference for the substrate promiscuity of A domains (L109).

We have added the Ref. 21 which demonstrated the substrate promiscuity of A domain from the hybrid enzyme AnATPKS with L-phenylalanine and its analogues as substrates. (Hai, Y. et al. J. Nat. Prod. (2020) 83:593-600) in L108.

6) L114: Here and in other places, the authors write “**3** and **4**” or something similar. This could literally mean that both substrates were added at once. If they were added individually, it should be “**3** or **4**”.

Many thanks. We have revised “**3** and **4**” to “**3** or **4**” when they were added individually in the full text.

7) Please note that “covalenting” is not a word. Maybe it should be.

We have revised “covalenting with” to “covalently attached to”.

8) L169/170: Methionine residues usually oxidize from a thioether to a sulfoxide not to an alcohol.

We have revised the sentence as “M608-oxidation from the thioether to the

sulfoxide” in L167 and the corresponding structures in Fig. 4c.

9) For describing the engineered yeast strains (P10ff), it would help to give names to the important strains to be able to precisely refer back to them when another gene is added. For example “QL35-NAR” for the naringenin producer etc.

Thank you very much for your suggestion. For better understanding, new names as QL35-NAR, QL35-ISO, and QL35-ACA have been given for the naringenin, isorhamnetin, and acacetin producers, respectively. Corresponding description has been rephrased in the text and Fig. 6c.

10) L240-244: This part continues to misrepresent what has been discovered in this study. The listed enzymes are examples where novel reaction mechanisms yielding novel structures have been discovered. This is not the novelty of this study. “Special enzyme” is a very blurry description.

Many thanks for your suggestion. We have deleted the description of examples of novel mechanism by genome mining strategy.

11) L263: Ref. 38 is not a good reference to support the sometimes promiscuous role of A-domains and there is a typo in the reference title.

Two publications revealing the substrate promiscuity of NRPS A domains have been cited (Crawford, J.M. et al., *Org. Lett.* (2011) 13: 5144-5147 and (Zhu, M. et al., *ACS. Chem. Biol.* (2019) 14: 256-265).

12) The attached version is a first step towards a readable text. For instance, the grammatical tenses are not ideally or wrongly used throughout the text and have not been carefully corrected in the attached version. More work is needed by a professional editor.

Thank you very much for the suggestions. Our co-author Prof. Shu-Ming Li and I have done intensive edits for the language problem including the grammar, tense mistakes, etc throughout the main text and method. Please see the revised version.

13) P11: What is “small fermentation”? Explicit volumes would be helpful.

We have revised “small fermentation” to “100 mL-YPD medium”.

Reviewer #3

The revisions made by the authors address most of my concerns and have substantially improved the manuscript. A few issues still remain (numbered according to the response letter), and there are still some unaddressed concerns regarding the novelty of the presented work:

Remarks:

1) Comment 1, Title: To my understanding FnsA has not been engineered to produce any flavonoid other than naringenin. The enzyme has been expressed in an engineered strain that allows for the subsequent conversion of naringenin into isorhamnetin and acacetin. Therefore, the title could be amended to “A fungal NRPS-PKS enzyme catalyzes formation of the flavonoid naringenin” to be less ambiguous. The current title still seems to be somewhat overstating the case

Thank you very much for your suggestion. We have already revised the title as “A fungal NRPS-PKS enzyme catalyses formation of the flavonoid naringenin”.

2) Comment 4, Line 29: The authors may want to revise the sentence. Biosynthetic pathways are refactored, not compounds.

We have rephrased the sentence as “Refactoring biosynthetic pathway in yeast with involvement of *fnsA* provides an alternative approach for the production of flavonoids such as isorhamnetin and acacetin.”

3) Comment 5, Line 53: The text needs to be revised. CHS and 4CL from *Alternaria* sp. MG1 have been heterologously expressed in *S. cerevisiae*, and their functions have been experimentally validated (<https://pubmed.ncbi.nlm.nih.gov/33453581/>). Furthermore, the text in the

discussion (lines 238 and following) needs to be revised accordingly.

We have changed the text to “Notably, only two examples of 4CL- and CHS-like proteins identified from endophytic fungi *Alternaria* sp. MG1¹⁷ and *Phomopsis liquidambaris*¹⁵ have been reported for the catalysation of naringenin and its derivatives. The biosynthesis of fungal-derived flavonoids is still rarely studied.”

The description has been revised to “However, little is known about the genes or pathways involved in their biosynthesis.”

4) Comment 15, Line 226: Please explain how the reference supports the claim that naringenin has attracted attention for decades. The cited paper (“Glycosylation Modification Enhances (2 S)-Naringenin Production in *Saccharomyces cerevisiae*”) was published in June 2022 and does not represent decades of attention.

Sorry for the confusion. To clarify this, we have changed the cited reference and revised the sentence as “It has attracted great attention for its biosynthesis and product efficiency ^{8,35,36} (Leonard, E. et al. *Appl. Environ. Microbiol.* (2007) 73:3877-3886; Sun, J. et al. *Metab. Eng.* (2022) 70: 143-154; Zang, Y. et al. *J. Agric. Food Chem.* (2019) 67: 13430-13436).

5) Comment 21, Line 270ff: As requested earlier, in the manuscript please provide more context regarding the concentrations of naringenin that have previously been achieved by expression of 4CL, CHS, and CHI in yeast. It is not sufficient to discuss this in the response to the reviewers. Furthermore, according to Wang et al., acacetin levels of 20.3 mg/L were obtained, which would be substantially higher than the 10.4 mg/L obtained by the authors. Where does the stated comparative titre of 4.2 mg/L come from?

We have revised the description according to your comments as follows:

“Although the yield of naringenin is relatively low in comparison with 1,184 mg·L⁻¹ from optimized *S. cerevisiae* strain by expression of 4CL, CHS, and

CHI⁴⁸, the refactoring isorhamnetin biosynthetic pathway has not been reported prior to this study. Furthermore, according to Wang et al³³, the acacetin product yield achieved to 20.3 mg·L⁻¹ under co-cultivation of three engineered strains, but merely up to 2.7 mg·L⁻¹ under mono-cultivation. In our case, the titer of acacetin attained 10.4 mg·L⁻¹ under our cultural condition.”

We deleted the wrong description of 4.2 mg/L.

REVIEWERS' COMMENTS

Reviewer #1 (Remarks to the Author):

The authors are once more congratulated for the interesting work they have done. The manuscript has been considerably improved and is now ready for publication.

Only a few minor edits are still necessary:

L23: "for elongation, respectively"; L54: "have been reported to catalyse formation of naringenin ..."; L108-109: "... promiscuity of A domains ... pCA (3) or simple molecules such as ..."; L120: "To determine whether production of 2 ..."; L141: "For further confirmation, ..."; L145: "the FnsA A domain"; L148: "the PKS portion"; L151: "the T domain"; L155: "N-terminally His6-tagged"; L171: "substrate specificity"; L183: "of type I, II, and III PKSs"; L213: space before "to construct"; L219: "space after NAR"; L224: "was increased 2.1-fold"; L252: "NRPS assembly lines"; L254: "that A domains can"; L268: "FnsA's one-protein synthesis of naringenin" (there are several enzymes enchainned on one protein); L309: "FnsA was used"; L358 delete "was"; L426: "The cultures were incubated at 28°C for 48 h and extracted twice with one volume of EtOAc."; L503: "Purity of the protein was confirmed by SDS-PAGE."; L506: "malony CoA" (small case); L515: "cut out"

L529-550: There are several instances of "The cassettes ... were conducted" or similar". That's grammatically not possible. One could say, for instance, "Gene disruption of ARO7 in TYHJ16 was conducted using a knock-out cassette according to the published protocol." Please correct all instances accordingly.

L539: "The gDNA ... was extracted"; L567ff: "UV absorption was recorded at 280 nm."; L590-592: please revise, meaning is completely unclear to the reviewer.

Figure 4 might better fit into the supporting information because it is quite technical, but this is entirely up to the authors.

Reviewer #3 (Remarks to the Author):

The authors have addressed my remaining concerns convincingly.

Reviewer #1 (Remarks to the Author):

The authors are once more congratulated for the interesting work they have done. The manuscript has been considerably improved and is now ready for publication.

Thank you very much again for the positive comments and very useful suggestions.

Only a few minor edits are still necessary:

L23: “for elongation, respectively”; L54: “have been reported to catalyse formation of naringenin ...”; L108-109: “... promiscuity of A domains ... pCA (3) or simple molecules such as ...”; L120: “To determine whether production of 2 ...”; L141: “For further confirmation, ...”; L145: “the FnsA A domain”; L148: “the PKS portion”; L151: “the T domain”; L155: “N-terminally His6-tagged”; L171: “substrate specificity”; L183: “of type I, II, and III PKSs”; L213: space before “to construct”; L219: “space after NAR”; L224: “was increased 2.1-fold”; L252: “NRPS assembly lines”; L254: “that A domains can”; L268: “FnsA’s one-protein synthesis of naringenin” (there are several enzymes enchainned on one protein); L309: “FnsA was used”; L358 delete “was”; L426: “The cultures were incubated at 28°C for 48 h and extracted twice with one volume of EtOAc.”; L503: “Purity of the protein was confirmed by SDS-PAGE.”; L506: “malony CoA” (small case); L515: “cut out”

We have corrected all issues you pointed out above in the revised manuscript.

L529-550: There are several instances of “The cassettes ... were conducted” or similar”. That’s grammatically not possible. One could say, for instance, “Gene disruption of ARO7 in TYHJ16 was conducted using a knock-out cassette according to the published protocol.” Please correct all instances accordingly.

Thanks. We have corrected the sentences throughout the text with highlights such as “Gene expression of *ubiC* and *aroL* in *S. cerevisiae* BJ5464-NpgA was

conducted using an integration cassette according to the described protocol”, “Gene disruption of *ARO7* in *S. cerevisiae* TYHJ16 was conducted using a knock-out cassette”, “Gene disruption of *TRP3* in *S. cerevisiae* TYHJ16 was conducted using a knock-out cassette”, and “Integration of isorhamnetin or acacetin pathways in QL35 was conducted using corresponding integration cassettes and gRNA vector pYHJ60, respectively”.

L539: “The gDNA ... was extracted”; L567ff: “UV absorption was recorded at 280 nm.”;

Done.

L590-592: please revise, meaning is completely unclear to the reviewer.

L590-592: We have rephrased the sentence to “The retrieval and identification of protein were performed by using the SEQUEST HT search engine of Thermo Proteome Discoverer (1.4.0.288) from the database (20210816YY0072-contaminations_m.fasata)”.

Figure 4 might better fit into the supporting information because it is quite technical, but this is entirely up to the authors.

Thank you for the suggestion. For better understanding the results of LC-MS/MS, we retained Figure 4 in the main text.

Reviewer #3 (Remarks to the Author):

The authors have addressed my remaining concerns convincingly.